

# Biology of tiny animals: three new species of minute salamanders (Plethodontidae: *Thorius*) from Oaxaca, Mexico

Gabriela Parra-Olea[1], Sean M. Rovito[2], Mario García-París[3], Jessica A. Maisano[4], David B. Wake[5] and James Hanken[6]

[1] Instituto de Biología, Universidad Nacional Autónoma de México, Mexico City, Mexico
[2] Unidad de Genómica Avanzada (Langebio), CINVESTAV, Irapuato, Guanajuato, Mexico
[3] Departamento de Biodiversidad y Biologia Evolutiva, Museo Nacional de Ciencias Naturales (MNCN-CSIC), Madrid, Spain
[4] Jackson School of Geosciences, University of Texas at Austin, Austin, Texas, United States
[5] Department of Integrative Biology and Museum of Vertebrate Zoology, University of California, Berkeley, California, United States
[6] Department of Organismic and Evolutionary Biology and Museum of Comparative Zoology, Harvard University, Cambridge, Massachusetts, United States

Corresponding authors
Gabriela Parra-Olea,
gparra@ib.unam.mx
James Hanken,
hanken@oeb.harvard.edu

## ABSTRACT

We describe three new species of minute salamanders, genus *Thorius*, from the Sierra Madre del Sur of Oaxaca, Mexico. Until now only a single species, *T. minutissimus*, has been reported from this region, although molecular data have long shown extensive genetic differentiation among geographically disjunct populations. Adult *Thorius pinicola* sp. nov., *T. longicaudus* sp. nov., and *T. tlaxiacus* sp. nov. are larger than *T. minutissimus* and possess elliptical rather than oval nostrils; *T. pinicola* and *T. longicaudus* also have longer tails. All three new species occur west of the range of *T. minutissimus*, which has the easternmost distribution of any member of the genus. The new species are distinguished from each other and from other named *Thorius* in Oaxaca by a combination of adult body size, external morphology and osteology, and by protein characters (allozymes) and differences in DNA sequences. In addition, we redescribe *T. minutissimus* and a related species, *T. narisovalis*, to further clarify the taxonomic status of Oaxacan populations and to facilitate future studies of the remaining genetically differentiated *Thorius* that cannot be satisfactorily assigned to any named species. Populations of all five species considered here appear to have declined dramatically over the last one or two decades and live specimens are difficult to find in nature. *Thorius* may be the most endangered genus of amphibians in the world. All species may go extinct before the end of this century.

## INTRODUCTION

The smallest salamanders in Mexico, members of the family Plethodontidae, belong to the genus *Thorius Cope, 1869*. Taxonomy of *Thorius* has proven difficult because of their small size and general morphological similarity, especially externally, but once the taxa

are sorted by using molecular characters, morphological features that distinguish species are often apparent. Indeed, our recent overview of the genus argues that *Thorius*, instead of comprising a proliferation of cryptic taxa, has undergone an adaptive radiation in miniature (*Rovito et al., 2013*). Characters used to distinguish species in other plethodontid genera typically include the number of trunk vertebrae, external color pattern, numbers of premaxillary, maxillary or vomerine teeth, relative limb length, and characteristics of the manus, pes and digits. All *Thorius*, however, have 14 trunk vertebrae and reduced limbs, and in most species the digits are poorly formed and syndactylous and maxillary teeth are absent. Furthermore, while there is little consistent variation in external coloration among most species, such comparisons are confounded by extensive individual variation both within and among conspecific populations.

By 1980, 10 formal names were available for populations found in four states—Guerrero, Oaxaca, Puebla and Veracruz. Population sizes were characteristically dense at that time, especially in mountains along the southeastern margin of the Mexican plateau. Sympatric species pairs were diagnosed mainly by small differences in adult body size and in size and shape of the external nares, which varied from small and round, to large and oval, to very large and elliptical (e.g., *Taylor, 1940*). Taxonomy, however, was problematic overall. It was difficult if not impossible to confidently associate names with most populations, and there was a general sense that many additional species remained undescribed.

A breakthrough came with the application of electrophoretic methods to study proteins. *Hanken (1983a)* assessed patterns of protein (allozyme) variation among nearly 70 populations from throughout the range. He found numerous additional instances of sympatry, including, in several cases, three species. Once sympatric species were detected, usually by the presence of many fixed genetic differences, specimens from a given locality could be sorted unequivocally. This, in turn, revealed reliable, albeit subtle characters from external morphology, osteology and/or dentition that differentiated species. Subsequent taxonomic studies were regionally focused: northern Oaxaca (*Hanken & Wake, 1994*; *Hanken & Wake, 2001*; *Wake et al., 2012*); Veracruz and Puebla (*Hanken & Wake, 1998*); and Guerrero (*Hanken, Wake & Freeman, 1999*; *Campbell et al., 2014*). This work led to the discovery and description of several new species; the number of valid, named taxa in *Thorius* more than doubled to the current 26.

Hanken's allozyme study revealed that most species of *Thorius* have very small geographic ranges. Indeed, many species are endemic to narrow altitudinal bands on a single mountain (e.g., *Hanken & Wake, 1994*). Hanken had successfully obtained topotypic samples for most named species, so many of the outstanding taxonomic issues could be resolved. Many new species were identified initially by allozymic characters that differentiate sympatric congeners, but a few were described in the absence of such data based on their extralimital distributions combined with discrete morphological differences from geographically adjacent species that were identified by molecular traits. Each new taxon described without genetic data was known at the time from fewer than five specimens collected in atypical habitats (usually, low elevations) (*Hanken & Wake, 1994*; *Hanken, Wake & Freeman, 1999*).

Until recently all attempts had failed to obtain topotypic specimens of *Thorius minutissimus Taylor, 1949*, the southernmost and easternmost taxon in the genus (type locality: Santo Tomás Teipan in the Sierra Madre del Sur of southeastern Oaxaca; Fig. 1). *Hanken (1983a)* reported extensive genetic differentiation among populations from southern Oaxaca, which presented repeated instances of sympatric species. Only a single species name was available for *Thorius* from the entire area, and it was uncertain which, if any, of Hanken's samples were assignable to *T. minutissimus*. Furthermore, all other Oaxacan populations to the north and west could be eliminated as close relatives based either on allozymes, morphology, or both. Ultimately, Hanken selected a population near Sola de Vega, a village in the Sierra Madre del Sur of Oaxaca about 115 km west of Santo Tomás Teipan, to represent *T. minutissimus*, and all subsequent literature using that species name refers solely to specimens from that locality (e.g., *Hanken, 1982*; *Hanken, 1983b*; *Hanken, 1984*). Nevertheless, while allozymic data and numerous instances of sympatry suggest the presence of several undescribed species in Hanken's samples, resolution of the taxonomic status of all *Thorius* from this region is not possible without definitive genetic data from topotypic *T. minutissimus*. Finally, following many unsuccessful attempts, two live adult *Thorius* were collected from Santo Tomás Teipan in early 2001, on the same trip that yielded another new species of plethodontid salamander from the region, *Bolitoglossa zapoteca* (*Parra-Olea, García-París & Wake, 2002*).

A recently published molecular phylogeny for *Thorius* enabled us to confirm the taxonomic distinctiveness of several species, including *T. minutissimus* from the type locality, which is related more closely to montane species from northern Oaxaca than to those from southern Oaxaca (*Rovito et al., 2013*). Hence, assignment of the Sola de Vega population to *T. minutissimus* (*Hanken, 1983a*) is incorrect. The molecular phylogeny, which is based on DNA sequence data from three mitochondrial genes (large subunit ribosomal RNA, 16S; cytochrome *b*, cyt *b*; and NADH dehydrogenase subunit 4, ND4) and one nuclear gene (RAG-1), also shows the presence of multiple lineages in southern Oaxaca that cannot be assigned to any named species. The phylogeny resolves three distinct clades, each supported by a posterior probability of 0.99 or 1.0.

In light of the above phylogeny, we here resolve several taxonomic issues that relate to the southeastern limits of the range of *Thorius*. We describe three new species and formally revise and supplement the original descriptions of *T. minutissimus* and *T. narisovalis Taylor, 1940*. These five species are differentiated from each other and from all other Oaxacan *Thorius* by a combination of external morphology, osteology, allozymic differences and/or phylogenetic analysis of DNA sequences. Some species are known from very few specimens, so comprehensive morphometric analyses are not possible for them. And while the new species are not always separable from one another or from congeners by discrete morphological characters, they occupy different positions in pairwise discriminant function analyses and several species pairs occur sympatrically with no evidence of interbreeding. All five species are assigned to clade 3 (*Rovito et al., 2013*).

The units of diversity we recognize herein are thought to represent populations or groups of populations on independent evolutionary trajectories (*Wiley, 1978*), as indicated by genetic data and/or as suggested by morphological data. They also reflect

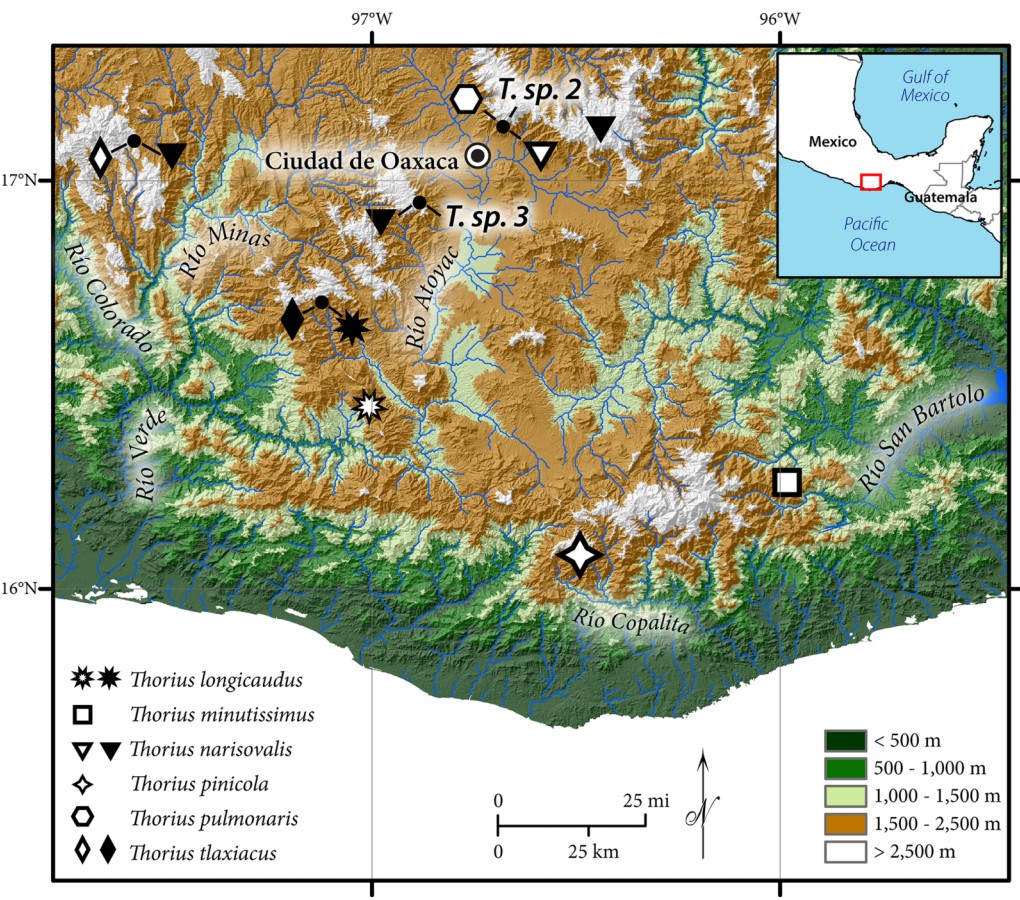

**Figure 1 Geographic distribution of *Thorius* in western and southern Oaxaca, Mexico.** Type localities of six named species are denoted by open symbols; additional localities are denoted by closed symbols. Known localities of *T.* sp. 2 and *T.* sp. 3, two unnamed Oaxacan species, are also shown (*Rovito et al., 2013*). Small closed circles denote four localities where two or three species are sympatric or nearly sympatric (from left to right): Heroica Ciudad de Tlaxiaco, San Vicente Lachixio, Zaachila and Cerro San Felipe.

relevant distributional and ecological information whenever possible (e.g., *Good & Wake, 1993*). Nearly all of the specimens analyzed in this study were collected more than 35 years ago because once-abundant natural populations of *Thorius* have declined dramatically; living specimens have become nearly impossible to find in nature. Most named species, including those described here, are highly endangered and are at serious risk of extinction.

## MATERIALS AND METHODS

Measurements were made of 7–10 adult males and 4–10 adult females of each new species and of *Thorius narisovalis*, the only well known and widely distributed species that occurs nearby. Only three adult specimens of *T. minutissimus* were measured: two recently collected females (IBH 23011–12) and one male collected in 1955 (MCZ 30869). Measurements were made with digital or dial calipers or a dissecting microscope fitted with an ocular micrometer. Standard length (SL) was measured from the anterior tip of

the snout to the posterior angle of the vent. Tail length (TL) was measured from the posterior angle of the vent to the tail tip. Limb interval (LI) equals the number of costal interspaces between the tips of appressed forelimbs and hind limbs, measured in one-half increments (e.g., 3, 4.5). Descriptions of relative limb and TL follow *Rovito et al. (2013)*: limb length—short (LI = 6–7), moderate (LI = 5–6), long (LI = 4–5) and very long (LI < 4); TL—very long (SL/TL < 0.8), long (SL/TL = 0.8–0.9), moderately long (SL/TL = 0.9–1.0), short (SL/TL = 1.0–1.2) and very short (SL/TL > 1.2).

Osteological descriptions are based primarily on examination of 20 cleared-and-stained adults of each species except *T. minutissimus* and *T. tlaxiacus* (1 and 0 specimens, respectively). In addition, an X-ray micro-computed tomography (μCT) scan was prepared from a single specimen of each species (see http://www.DigiMorph.org/), and as many as 10 additional specimens per species were digitally X-rayed to count caudal vertebrae. Whole-mount skeletal preparations were stained for bone and cartilage using alizarin red S and Alcian blue 8GX, respectively (*Klymkowsky & Hanken, 1991*). Cranial character states and mesopodial patterns are described and illustrated by *Hanken (1982)*, *Hanken (1984)*, *Hanken (1985)*, *Hanken & Wake (1994)*, *Hanken & Wake (1998)*, *Hanken & Wake (2001)* and *Hanken, Wake & Freeman (1999)*; see *Wake & Elias (1983)* for comparisons with other tropical genera. Fused distal carpals 1-2 and fused distal tarsals 1-2, which are synapomorphies of crown-group Urodela, equal the "basal commune" of other authors (*Shubin, Wake & Crawford, 1995*). Counts of presacral vertebrae do not include the first vertebra (atlas). Tooth counts are based on cleared-and-stained specimens except those for holotypes and the reference samples of *T. minutissimus* and *T. tlaxiacus*, which are ethanol-preserved; all ethanol-preserved specimens were examined for the presence of maxillary teeth. Numbers of vomerine teeth in each holotype are provided separately for right and left sides; these counts are summed for other individuals.

Comparisons are limited to Oaxacan members of clade 3 of *Rovito et al. (2013*, Fig. 2), which includes all five species considered herein, plus three additional Oaxacan species from clade 2 (*T. adelos, T. insperatus, T. smithi*; *Rovito et al., 2013*, Fig. 3). Institutional abbreviations are as listed in *Sabaj (2016)*.

Statistical analyses were performed using Statistica (v. 8) and R (*R Development Core Team, 2014*). We used linear discriminant function analysis (DFA), performed using the MASS package (*Venables & Ripley, 2002*), to evaluate the ability of morphological characters to differentiate species from their respective type localities. We included the three new species described here, as well as *T. narisovalis* and *T. minutissimus*, and based the analysis on eight log-transformed variables: SL, shoulder width, head length, head width, hind limb length, axilla-groin distance, foot width and the ratio of nostril dimensions (major axis/minor axis). Wilk's lambda was used to test for significance of differences among groups (species).

Animal use was approved by the University of California, Berkeley, IACUC protocol #R093-0205 issued to D.B.W. Collection of live salamanders in the field was authorized by the Secretaria de Recursos Naturales y del Medio Ambiente (SEMARNAT), Mexico, permit no. FAUT-0106 issued to GP-O.

The electronic version of this article in Portable Document Format (PDF) will represent a published work according to the International Commission on Zoological Nomenclature (ICZN), and hence the new names contained in the electronic version are effectively published under that Code from the electronic edition alone. This published work and the nomenclatural acts it contains have been registered in ZooBank, the online registration system for the ICZN. The ZooBank LSIDs (Life Science Identifiers) can be resolved and the associated information viewed through any standard web browser by appending the LSID to the prefix http://zoobank.org/. The LSID for this publication is: urn:lsid:zoobank.org:pub:83638F13-8A23-40F1-9992-100246084196. The online version of this work is archived and available from the following digital repositories: PeerJ, PubMed Central and CLOCKSS.

## RESULTS

Means and standard deviations of all external measurements and tooth counts for adults of five species of *Thorius* from their respective type localities are shown in Table 1. Because all five type localities are geographically distinct, each species sample is from a different locality and no samples are sympatric with one another. Classification probabilities obtained from the linear DFA of eight variables assign 86% of the 66 specimens to the correct species (Table 2; Fig. 2). All specimens of *T. minutissimus* are correctly assigned; four specimens of *T. pinicola*, two each of *T. longicaudus* and *T. tlaxiacus*, and one of *T. narisovalis* are misclassified. Six of the eight variables passed a normality test. Repeating the DFA without the two non-normal variables (shoulder width and foot width) again yields high classification probabilities (70% or higher) for four of the five species. The probability of correct classification of *T. pinicola*, however, drops to 46% from 69%. In the original DFA, single specimens each of *T. longicaudus* and *T. tlaxiacus* occupy almost identical positions in the morphospace, as do single specimens each of *T. tlaxiacus* and *T. narisovalis* (Fig. 2). Although each of these species pairs may occur in sympatry, the particular specimens represented here were collected from different localities. Possible patterns of morphological divergence among species due to sympatry versus allopatry cannot be evaluated from these data.

### *Thorius pinicola*, new species
Pine-dwelling Minute Salamander
Figure 3

*Thorius* sp. nov.—*Mueller et al., 2004*: Table 2, Figs. 1 and 2.

*Thorius* sp. nov.—*Vieites et al., 2011*: Figs. 1–3.

*Thorius* sp.—*Wiens et al., 2007*: Fig. 2.

*Thorius* sp. 6.—*Rovito et al., 2013*.

*Holotype*: MVZ 185344, Mexico, Oaxaca, Miahuatlán District, Mexico Hwy. 175, 4.2 mi N (by road) San Miguel Suchixtepec, adult female, 16°7′11″N, 96°29′26″W, 2,700 m above sea level, 16 July 1976, J. F. Lynch and J. Hanken.

**Table 1 External measurements (in mm) and tooth counts for five species of *Thorius*.**

| Species | | SL | SW | HL | HW | AxGr | LI | HLL | FW | Nma/Nmi | Nma | Nmi | PT | VT |
|---|---|---|---|---|---|---|---|---|---|---|---|---|---|---|
| *T. pinicola* | $\bar{x}$ | 26.5 | 2.5 | 4.4 | 3.2 | 15.4 | 5.6 | 4.3 | 1.2 | 1.7 | 0.68 | 0.41 | 1.4 | 5.7 |
| F | sd | 0.88 | 0.2 | 0.2 | 0.2 | 1.0 | 0.7 | 0.3 | 0.1 | 0.3 | 0.09 | 0.06 | 0.8 | 1.4 |
| | Min | 25.5 | 2.3 | 3.9 | 3.0 | 13.7 | 5.0 | 3.7 | 1.1 | 1.2 | 0.6 | 0.3 | 0 | 3 |
| | Max | 28.2 | 3.1 | 4.9 | 3.5 | 16.7 | 7.0 | 4.9 | 1.4 | 2.3 | 0.8 | 0.5 | 3 | 8 |
| *T. pinicola* | $\bar{x}$ | 25.7 | 2.5 | 4.6 | 3.0 | 14.2 | 5.3 | 4.5 | 1.1 | 1.7 | 0.76 | 0.45 | 1.5 | 4.8 |
| M | sd | 1.79 | 0.1 | 0.2 | 0.1 | 1.5 | 0.8 | 0.4 | 0.1 | 0.1 | 0.07 | 0.05 | 0.5 | 0.8 |
| | Min | 23.5 | 2.3 | 4.4 | 2.9 | 12.4 | 4.0 | 3.6 | 1.1 | 1.5 | 0.6 | 0.4 | 1 | 4 |
| | Max | 29.6 | 2.6 | 5.1 | 3.2 | 17.4 | 6.0 | 4.7 | 1.4 | 1.6 | 0.8 | 0.5 | 2 | 6 |
| *T. longicaudus* | $\bar{x}$ | 25.5 | 2.8 | 4.3 | 3.1 | 14.4 | 5.5 | 4.2 | 1.2 | 1.8 | 0.6 | 0.4 | 1.8 | 7.9 |
| F | Sd | 1.0 | 0.1 | 0.2 | 0.1 | 0.5 | 0.4 | 0.3 | 0.07 | 0.3 | 0.05 | 0.06 | 1.0 | 1.3 |
| | Min | 24.4 | 2.6 | 4.1 | 2.9 | 13.6 | 5.0 | 3.7 | 1.0 | 1.4 | 0.6 | 0.3 | 0 | 6 |
| | Max | 27.7 | 3.2 | 4.8 | 3.3 | 15.8 | 6.0 | 4.9 | 1.3 | 2.3 | 0.7 | 0.4 | 4 | 10 |
| *T. longicaudus* | $\bar{x}$ | 25.0 | 2.8 | 4.3 | 3.1 | 14.1 | 5.3 | 4.6 | 1.2 | 1.8 | 0.7 | 0.4 | 1.1 | 7.3 |
| M | sd | 1.4 | 0.2 | 0.2 | 0.1 | 0.8 | 0.3 | 0.4 | 0.0 | 0.2 | 0.1 | 0.0 | 0.3 | 1.7 |
| | Min | 23.6 | 2.6 | 4.0 | 3.0 | 13.2 | 5.0 | 4.3 | 1.1 | 1.5 | 0.6 | 0.4 | 1 | 5 |
| | Max | 28.3 | 3.1 | 4.5 | 3.4 | 16.0 | 5.5 | 5.0 | 1.2 | 1.75 | 0.8 | 0.4 | 2 | 10 |
| *T. tlaxiacus* | $\bar{x}$ | 27.7 | 2.5 | 4.8 | 3.5 | 16.2 | 5.3 | 4.6 | 1.4 | 2.3 | 0.6 | 0.3 | 0.5 | 6.3 |
| F | sd | 3.6 | 0.3 | 0.4 | 0.3 | 2.0 | 0.9 | 0.6 | 0.2 | 0.3 | 0.06 | 0.06 | 0.6 | 1.7 |
| | Min | 22.6 | 2.2 | 4.4 | 3.1 | 13.7 | 4.5 | 3.7 | 1.2 | 2.0 | 0.5 | 0.2 | 0 | 4 |
| | Max | 31.0 | 2.7 | 5.3 | 3.8 | 18.4 | 6.0 | 4.9 | 1.5 | 2.5 | 0.6 | 0.3 | 1 | 8 |
| *T. tlaxiacus* | $\bar{x}$ | 28.0 | 2.7 | 4.8 | 3.4 | 15.5 | 4.3 | 4.8 | 1.3 | 2.1 | 0.6 | 0.3 | 1.3 | 4.9 |
| M | Sd | 3.2 | 0.4 | 0.5 | 0.3 | 2.2 | 0.6 | 0.5 | 0.2 | 0.3 | 0.09 | 0.04 | 0.8 | 0.9 |
| | Min | 21.1 | 2.0 | 3.8 | 2.7 | 11.1 | 3.5 | 3.8 | 1.0 | 1.7 | 0.5 | 0.2 | 0 | 4 |
| | Max | 30.2 | 3.0 | 5.1 | 3.7 | 17.6 | 5.0 | 5.2 | 1.6 | 2.5 | 0.7 | 0.3 | 2 | 6 |
| *T. narisovalis* | $\bar{x}$ | 27.8 | 2.7 | 4.7 | 3.5 | 15.8 | 5.7 | 4.2 | 1.3 | 1.4 | 0.4 | 0.3 | 1.2 | 4.7 |
| F | Sd | 1.3 | 0.2 | 0.3 | 0.2 | 1.2 | 0.4 | 0.3 | 0.1 | 0.2 | 0.0 | 0.1 | 0.9 | 1.1 |
| | Min | 26.2 | 2.6 | 4.3 | 3.2 | 14.3 | 5.0 | 3.8 | 1.2 | 1.25 | 0.4 | 0.3 | 0 | 3 |
| | Max | 29.9 | 3.0 | 5.1 | 3.7 | 17.5 | 6.5 | 4.6 | 1.5 | 2.0 | 0.5 | 0.4 | 3 | 7 |
| *T. narisovalis* | $\bar{x}$ | 25.2 | 2.5 | 4.6 | 3.3 | 14.2 | 5.0 | 4.3 | 1.3 | 1.4 | 0.4 | 0.3 | 0.5 | 4.3 |
| M | sd | 1.7 | 0.2 | 0.3 | 0.2 | 1.0 | 0.4 | 0.3 | 0.1 | 0.2 | 0.1 | 0.0 | 0.7 | 1.4 |
| | Min | 22.2 | 2.2 | 4.4 | 3.1 | 12.5 | 4.5 | 3.8 | 1.2 | 1.0 | 0.3 | 0.3 | 0 | 2 |
| | Max | 28.4 | 2.9 | 5.0 | 3.5 | 15.8 | 5.5 | 4.8 | 1.5 | 1.67 | 0.5 | 0.3 | 2 | 7 |
| *T. minutissimus* | $\bar{x}$ | 23.0 | 2.2 | 4.1 | 2.9 | 13.4 | 6.0 | 4.2 | 1.0 | 1.3 | 0.65 | 0.5 | 1.0 | 7.0 |
| F | sd | 0.9 | 0.1 | 0.1 | 0.1 | 0.5 | 0.7 | 0.3 | 0.1 | 0.1 | 0.04 | 0.03 | 0.0 | 2.8 |
| | Min | 22.3 | 2.1 | 4.0 | 2.8 | 13.0 | 5.5 | 4.0 | 0.9 | 1.2 | 0.62 | 0.48 | 1 | 5 |
| | Max | 23.6 | 2.2 | 4.2 | 3.0 | 13.7 | 6.5 | 4.4 | 1.0 | 1.4 | 0.67 | 0.52 | 1 | 9 |
| *T. minutissimus* | $\bar{x}$ | 19.8 | 1.8 | 4.0 | 2.9 | 10.5 | 5.0 | 3.8 | 0.74 | 1.2 | 0.48 | 0.4 | 2 | 7 |
| M | sd | – | – | – | – | – | – | – | – | – | – | – | – | – |
| | Min | – | – | – | – | – | – | – | – | – | – | – | – | – |
| | Max | – | – | – | – | – | – | – | – | – | – | – | – | – |

**Notes:**
$\bar{x}$, Means; sd, standard deviations; Min, minimum values and Max, maximum values are provided for M, adult males and F, adult females from the respective type localities.
Additional abbreviations: SL, snout-vent length; SW, shoulder width; HL, head length; HW, head width; AxGr, axilla-groin; LI, limb interval; HLL, hind limb length; FW, foot width; Nma, nostril major axis; Nmi, nostril minor axis; PT, premaxillary teeth; and VT, vomerine teeth.

**Table 2 Classification matrix obtained from the linear discriminant-function analysis of eight log-transformed morphological variables.** Each row depicts the predicted classification of specimens of a given species from its respective type locality. Fifty-seven of 66 specimens (86.4%) were correctly assigned to their respective species.

| Species | n | Percent classified correctly | *T. pinicola* | *T. longicaudus* | *T. tlaxiacus* | *T. narisovalis* | *T. minutissimus* |
|---|---|---|---|---|---|---|---|
| *T. pinicola* | 13 | 69.2 | 9 | 3 | 1 | 0 | 0 |
| *T. longicaudus* | 20 | 90.0 | 1 | 18 | 1 | 0 | 0 |
| *T. tlaxiacus* | 11 | 81.8 | 1 | 1 | 9 | 0 | 0 |
| *T. narisovalis* | 19 | 94.7 | 0 | 0 | 1 | 18 | 0 |
| *T. minutissumus* | 3 | 100 | 0 | 0 | 0 | 0 | 3 |

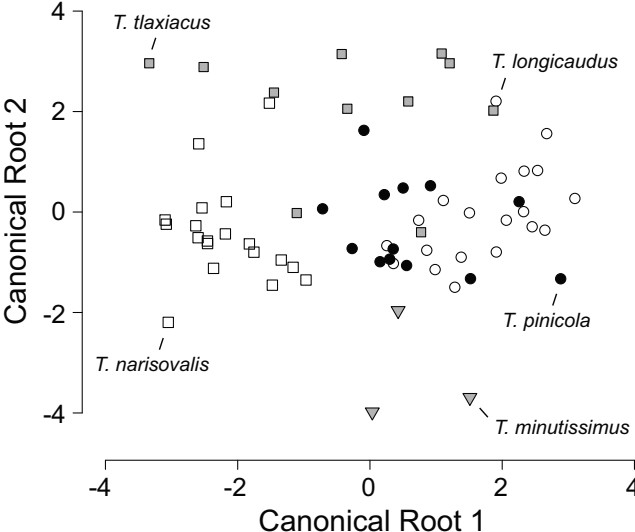

**Figure 2 Scatterplot of canonical scores (root 2 vs. root 1) generated by the discriminant function analysis of eight morphological variables in five species of *Thorius* from their respective type localities.** The analysis correctly assigns 86% of the 66 specimens to their respective species.

**Paratypes:** All from Oaxaca, Mexico: MVZ 185337–43 (seven specimens), 185345–48 (four specimens), 187146–60 (15 specimens) and 231444–46 (three specimens), same data as the holotype; MVZ 185325–36 (12 specimens) and 187141–45 (five specimens), Mexico Hwy. 175, 7.7 mi N (by road) San Miguel Suchixtepec, 16°8′57″N, 96°30′0″W, 2,490 m, 16 July 1976, J. F. Lynch and J. Hanken; MCZ A-136429 and IBH 13995, 13997, 1.7 km N (by road) San Miguel Suchixtepec, 16°06′20.4″N, 96°28′9.6″W, 2,630 m, 25 January 2001, G. Parra-Olea, M. García-París, J. Hanken and T. Hsieh; MZFC 16089, 4.8 mi NE (by road) Díaz Ordaz, 16°04′57″N, 96°23′41″W, 3,000 m, 23 September 2001, J. A. Campbell; MZFC 16131–33 (three specimens), Carretera La Venta-Cerro Nevería, 16°11′43″N, 96°21′56″W, 2,870–2,995 m, 1 October 2001, J. A. Campbell; MZFC 21789, Sierra Miahuatlán, 16°11.759′N, 96°21.977′W, 2,943 m, 1 October 2001, J. A. Campbell.

**Diagnosis:** Distinguished from other species of *Thorius* by the following combination of characters: (1) large size (SL exceeds 23 mm in males and 25 mm in females); (2) moderately short limbs; (3) long tail; (4) elongated, elliptical nostrils; (5) no maxillary teeth; and (6) few vomerine teeth (6 or fewer in males and 8 or fewer in females).

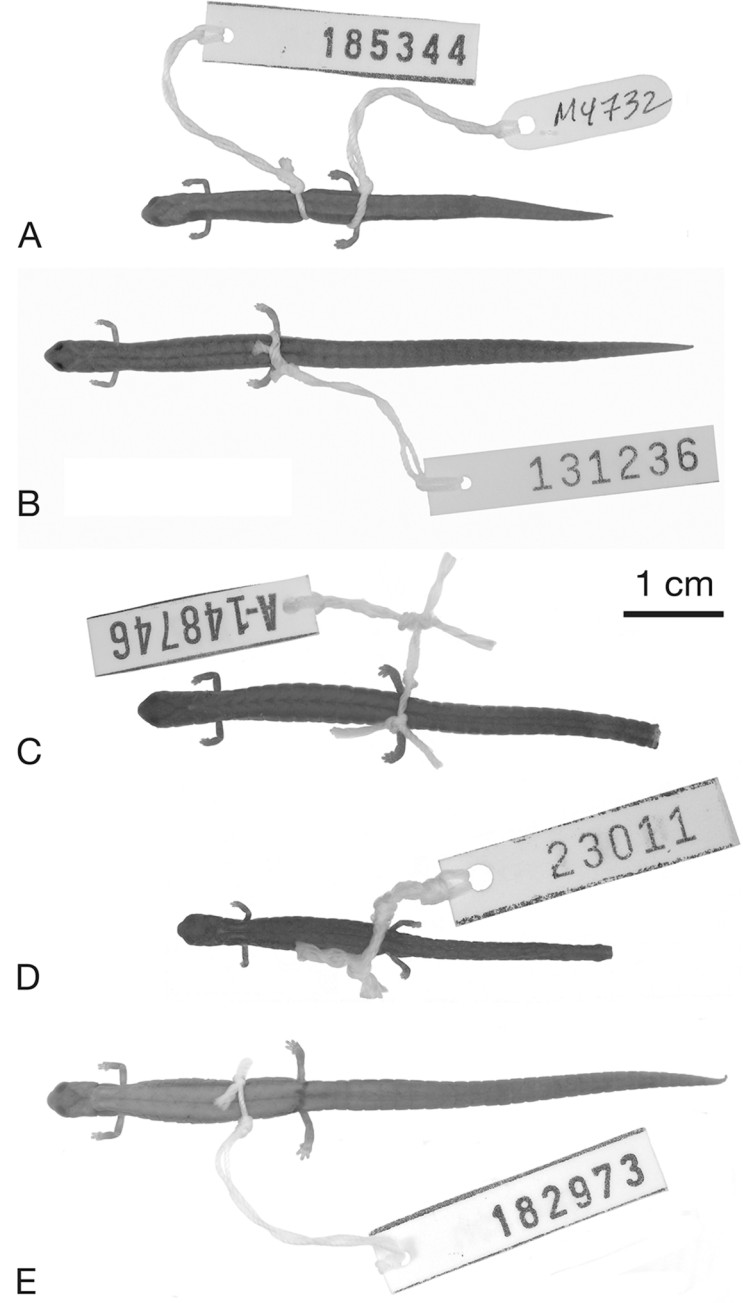

**Figure 3 Holotypes and referred specimens of five species of *Thorius* from Oaxaca, Mexico.**
(A) Holotype of *T. pinicola*, MVZ 185344, an adult female from 4.2 mi N of San Miguel Suchixtepec.
(B) Holotype of *T. longicaudus*, MCZ A-137819 (= MVZ 131236), an adult female from 19 km S of Sola
de Vega. (C) Holotype of *T. tlaxiacus*, MCZ A-148746, an adult female from 27.3 km SSE of Tlaxiaco.
The tail tip was removed for DNA sequencing. (D) *Thorius minutissimus*, IBH 23011, an adult female
from the type locality, 1.1 km W of Santo Tomás Teipan. The tail tip was removed for DNA sequencing.
(E) *Thorius narisovalis*, MVZ 182973, an adult female from Cerro San Felipe, 15 km W of La Cumbre.

***Comparisons:*** Adult *Thorius pinicola* are larger than *T. arboreus*, *T. insperatus*,
*T. minutissimus*, *T. papaloae* and *T. smithi*. The smallest-known adult *T. pinicola* is
23.5 mm SL and most adults, especially females, are larger than 25 mm. None of the other

species is known to exceed 23.6 mm and most adults, especially males, are smaller than 20 mm. *Thorius adelos, T. arboreus, T. insperatus, T. macdougalli* and *T. smithi* have relatively much longer limbs (LI < 4), while LI exceeds 4 in *T. pinicola*. The nostril in *T. pinicola* is large and elongated elliptical, whereas *T. narisovalis* has relatively small-to-moderate-sized, oval nostrils (occasionally round). The nostril is even more extremely distorted in *T. pulmonaris* and *T. tlaxiacus*, where it is prolate in shape. All *T. pinicola* lack maxillary teeth, which differentiates them from *T. adelos, T. aureus* and *T. smithi*. *Thorius pinicola* has fewer vomerine teeth (mean number in both males and females is between 4 and 6) than *T. longicaudus* (mean between 7 and 8) and *T. boreas* (mean between 9 and 10).

***Description of holotype***: Head width 3.1 mm; snout to gular fold (head length) 4.3 mm; head depth at posterior angle of jaw 2.2 mm; eyelid width 0.8 mm; eyelid length 1.6 mm; anterior rim of orbit to snout 0.9 mm; horizontal orbit diameter 1.3 mm; interorbital distance 1.3 mm; distance between corners of eyes 1.7 mm; distance separating external nares 0.9 mm; major axis of nostril 0.6 mm; minor axis of nostril 0.4 mm; snout projection beyond mandible 0.4 mm; snout to posterior angle of vent (SL) 25.5 mm; snout to anterior angle of vent 23.6 mm; snout to forelimb 6.9 mm; axilla to groin 14.8 mm; LI 6.0 costal interspaces; shoulder width 2.2 mm; TL 22.9 mm; tail width at base 2.5 mm; tail depth at base 2.5 mm; forelimb length (to tip of longest toe) 4.0 mm; hind limb length 4.5 mm; manus width 0.9 mm; pes width 1.2 mm. Numbers of teeth: premaxillary 2; maxillary 0; vomerine 3-4.

Overall ground color dark blackish-brown, darkest along flanks of trunk and tail (Fig. 3A). Obscure, brown dorsal stripe with indistinct borders begins on nape and extends onto proximal portion of tail. Venter pale brown, scattered white spots in gular region; ventral spots become indistinct in trunk. Limbs slightly paler brown than rest of animal; manus and pes even less densely pigmented. Costal grooves, gular fold and extension of fold onto neck are conspicuous because they lack pigment. Otherwise, no distinguishing marks. Parotoid gland prominent.

***Variation***: Mean adult SL 25.7 mm (range 23.5–29.6) in eight males, 26.5 mm (25.5–28.2) in nine females. Head narrow; SL 8.1 times head width in males (7.8–8.9) and 8.3 in females (7.5–8.5). Snouts bluntly pointed. Nostrils large and elliptical; ratio of major to minor axes 1.7 (1.5–1.8) in males, 1.7 (1.2–2.3) in females. Eyes moderately small, in a few specimens protrude slightly beyond jaw margins in dorsal view. Suborbital groove intersects lip on each side of head. Premaxillary teeth 1.5 (1–2) in adult males, 1.4 (0–3) in females. Maxillary teeth absent. Vomerine teeth 4.8 (4–6) in males, 5.7 (3–8) in females. Limbs moderately long; LI 5.3 (4.0–6.5) in males, 5.6 (5.0–7.0) in females. Manus and pes relatively well developed but narrow; foot width 1.1 mm in males (1.1–1.4) and 1.2 mm in females (1.1–1.4). Digits 1 and 4 (manus) and 1 and 5 (pes) short, almost completely fused to the neighboring digits; central digits relatively long, with rounded tips. Fingers, in order of decreasing length, 3-2-4-1; toes 3-4-2-5-1. Tail long and tapered; SL divided by TL 0.82 (0.75–0.90) in five males, 0.81 (0.61–1.11) in seven females. Mental gland round and relatively prominent in most adult males (maximum dimensions: 1.3 mm

wide, 1.3 mm long). Postiliac gland small, pale, relatively inconspicuous externally. Parotoid glands evident in most specimens, but less so in a few others.

*Coloration in life*: Ground color on flanks black suffused with fine white speckling; broad brassy copper dorsal band etched with thin black lines; band with occasional dark blotches or flecks but lacks conspicuous chevrons; venter pale with light speckling (J. Hanken field notes, 16 July 1976 and 25 January 2001; IBH 13995, 13997, MCZ A-136429, MVZ 185325–48, 187141–60, 231444–46; Fig. 4A).

*Coloration in preservative*: A relatively dark species, although coloration has lightened considerably in older preserved specimens. There is a more-or-less obscure dorsal band, and the palest bands have a herringbone pattern mid-dorsally. The dorsal band is more prominent in life. The venter is dark, but paler than the flanks; the underside of the tail is especially pale. The gular region is covered with numerous white spots. Many individuals have a pale nuchal spot; some have a pair of pale streaks over the shoulders.

*Osteology*: Skull delicate (Figs. 5A–5C). Many individual bones thin, with frequent right-left asymmetry in articulation between adjacent bones, especially anteriorly. Rostral portion of skull also shows modest sexual dimorphism involving premaxillary, maxillary, nasal and prefrontal bones, which typically are more robust and articulate more extensively in females. Contact between ascending processes of unpaired premaxillary bone highly variable in both sexes. Processes separate in some specimens (character 1, state a), articulate in varying degrees or fused (states b–d). Dental process of premaxilla separate from maxilla in most males and some females (character 2, state a); bones overlap slightly in ventral view (state b) or articulate (state d) in remaining specimens, especially females. Premaxilla with teeth (character 8, state b). Nasal bones are highly variable, ranging from thin and rod-like at the posterior edge of the cartilaginous nasal capsule (character 3, state b) to slightly broader, extending somewhat anteriorly (state c). They are irregularly shaped in many specimens, consisting of a broad but thin dorsal part with an uneven anterior border, and a thin ventral part; the two parts are separate in at least two specimens. Nasal and maxilla do not contact in most males and some females (character 4, state a); they articulate in most females and in one male (state b) and are fused in one female (state c). Prefrontal separate from nasal in nearly all males and in most females (character 5, state b) and divided on one or both sides of several specimens (both sexes). Prefrontal articulates with nasal (state c) in remaining specimens. Prefrontal typically well separated from maxilla (character 6, state a), but in a few specimens extends posteriorly and ventrally beyond nasolacrimal foramen to approach or contact maxilla (state b). Septomaxillary bone absent (character 7, state a).

Presacral vertebrae 14; first vertebra (atlas) divided transversely in one specimen (MVZ 187142). Trunk vertebrae except last bear ribs, except two specimens with partial ribs on one or both sides of last vertebra. Mean number of caudal vertebrae 26.3 (range 21–30) in three males; one female has 32 vertebrae.

Limbs slender but well developed. Tibial spur present as inconspicuous attached crest in most specimens, but ranges from well developed to absent in a few others.

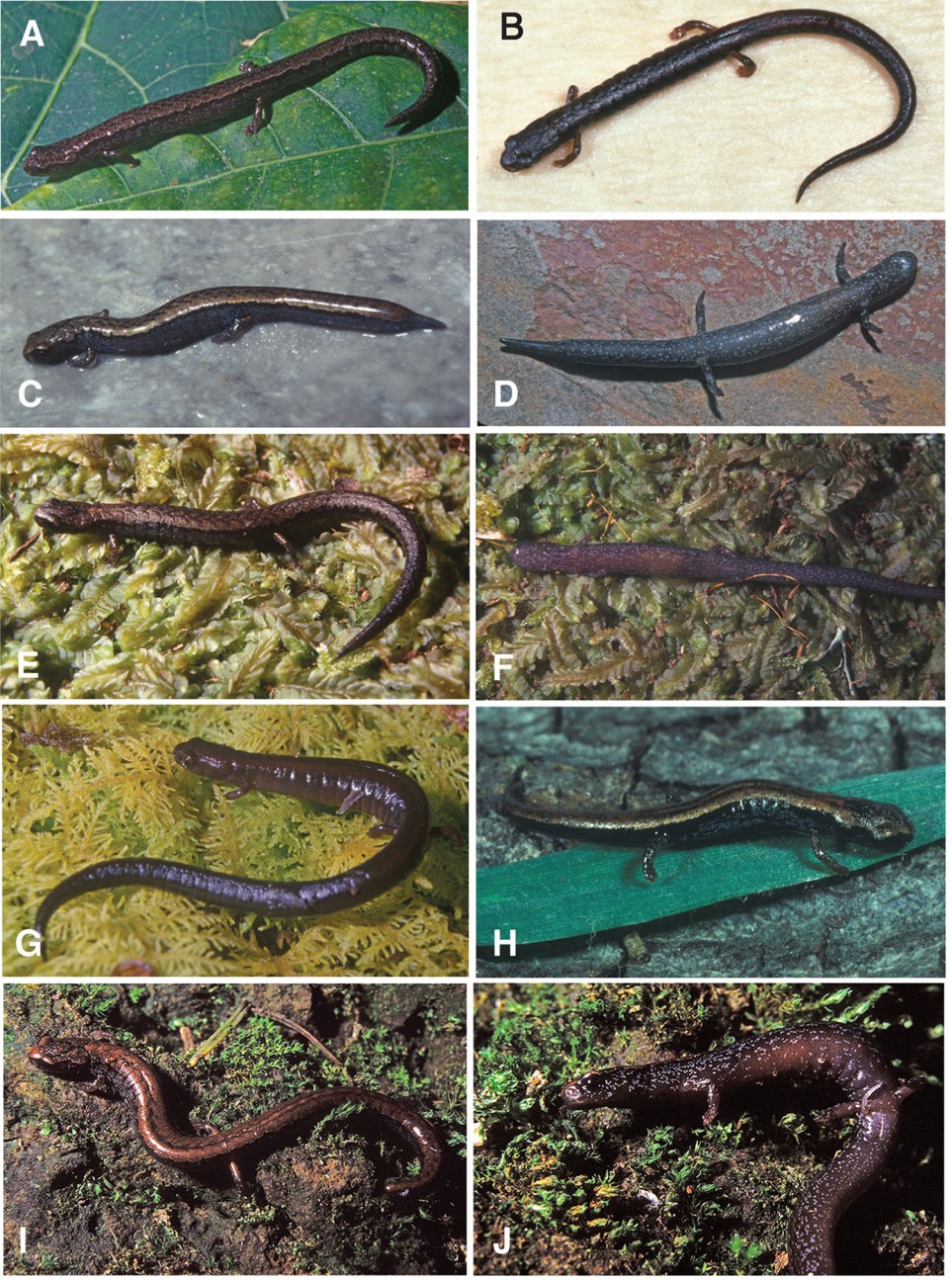

**Figure 4 Salamanders in life.** (A) *Thorius pinicola* from 1.7 km N of San Miguel Suchixtepec; MCZ A-136429. (B) *Thorius longicaudus* with complete tail from the type locality, 19 km S of Sola de Vega; museum number unavailable. (C, D) *Thorius longicaudus* with regenerating tail from the type locality, seen in dorsal and ventral views; IBH 13998. (E, F) *Thorius minutissimus* from the type locality, 1.1 km W of Santo Tomás Teipan, seen in dorsal and ventral views; IBH 23012. (G) *Thorius narisovalis* from Cerro San Felipe; IBH 14331. (H) Juvenile *T. narisovalis* from Cerro San Felipe lying on a blade of grass. (I, J) *Thorius tlaxiacus* from the type locality, 27.3 km SSE (by road) Tlaxiaco, seen in dorsal and ventrolateral views; MCZ 148746. (A, C–G, I and J) Photos by M. García-París and (B and H) J. Hanken.

Mesopodial morphology only slightly variable. Sole carpal pattern (I; 100% of limbs examined; Fig. 6A) contains six separate elements, with two derived states in relation to outgroup genera: fused intermedium plus ulnare, and fused distal carpal 4 plus centrale.

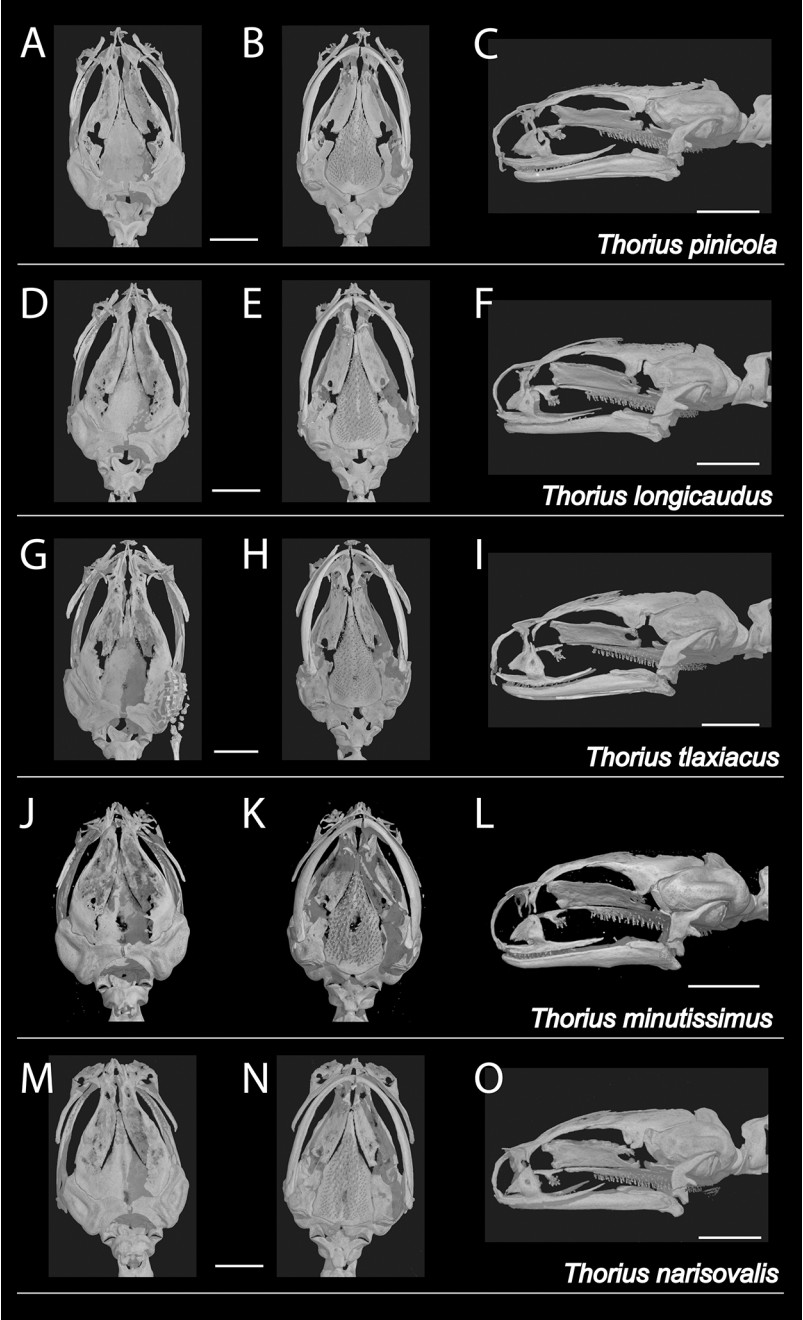

**Figure 5 X-ray micro-computed tomography (μCT) scans of adult skulls.** (A–C) *Thorius pinicola*, MCZ A-136429, paratype, male; (D–F) *T. longicaudus*, MCZ A-137819, holotype, female; (G–I) *T. tlaxiacus*, MVZ 183447, paratype, male; (J–L) *T. minutissimus*, IBH 23011, female; and (M–O) *T. narisovalis*, MVZ 162257, female. Each skull is shown in dorsal (left), ventral (middle) and left lateral views. The skeleton of the right hand is visible in G. Total length of each skull is only 3–4 mm; scale bar, 1 mm.

Modal tarsal pattern (I; 82%; Fig. 6B) contains eight separate elements, with one derived state in relation to outgroup genera: fused distal tarsals 4 and 5. Second tarsal pattern at moderate frequency (V; 18%; Fig. 6C) has one additional fusion relative to pattern I: intermedium plus fibulare.

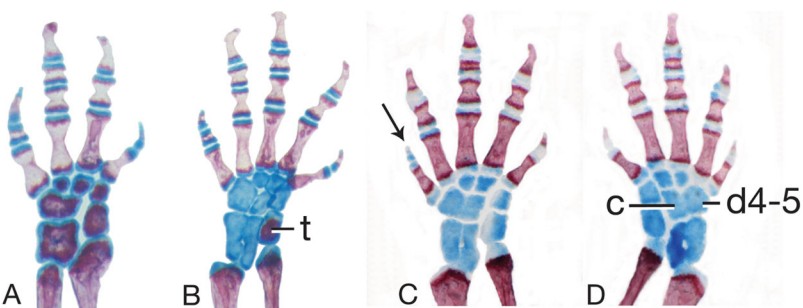

**Figure 6 Limb skeletal morphology and variation.** (A) The left hand of MVZ 186837, *Thorius long-icaudus*, displays carpal (wrist) pattern I, the predominant pattern in the genus, including all five species considered in the present study. (B) The left foot of MVZ 186824, *T. longicaudus*, displays tarsal (ankle) pattern I, which predominates in this species as well as in *T. pinicola, T. tlaxiacus* and *T. narisovalis*. (C, D) Left and right feet of IBH 23012, *T. minutissimus*, show bilateral asymmetry in tarsal pattern (V and VII, respectively). Distal tarsal 4–5 (d4–5) and the centrale (c) are fused in VII; they are separate, but overlapping, in V. Both patterns are otherwise rare in the genus. Note also the different phalangeal formulae between B (1-2-3-3-2) and C and D (1-2-3-2-1), which have a correspondingly short fifth toe (arrow). Cartilaginous (blue) tarsals and metatarsals in C and D indicate a subadult specimen. Cartilage is beginning to ossify in B (t, tibiale; red) and wrist elements are nearly fully ossified in A, indicating the onset of sexual maturity in these specimens. All limbs are shown in dorsal view.

Digital skeleton variable, especially in hind limb. Phalangeal formulae in manus 1-2-3-2 (92%) or 1-2-3-1 (8%), in pes: 1-2-3-3-2 (45%), 1-2-3-3-1 (45%); 1-2-3-2-1 (6%), 1-3-3-3-1 (3%). Limb bone epiphyses and mesopodial elements mineralized in several adults.

***Distribution and ecology***: *Thorius pinicola* is known from several localities along Mexico Hwy. 175, between 1.7 and 12.4 km north of the village of San Miguel Suchixtepec, Oaxaca, and also a few kilometers east of this region. These localities lie within a small mountain range that is a component of the Sierra Madre del Sur (Figs. 1 and 7A). Recorded elevations range from 2,490 to 2,700 m.

According to field notes of J. Hanken from 16 July 1976 (MVZ 185325–48, 187141–60 and 231444–46), the dominant natural habitat is pine-oak forest. All *Thorius* were taken in terrestrial habitats under charred fallen logs or in adjacent litter and pine needles. According to notes from 25 January 2001 (IBH 13995, 13997 and MCZ A-136429), the locality is a wooded slope extending to the ridgeline of surrounding hills. It is dominated by tall, slender pines with an understory of oak, madrone and small shrubs, but the nearby areas have been largely cleared of natural vegetation. Much logging activity has left the slopes littered with fallen logs. The forest, with several inches of leaf litter (pine needles) was dry to ground level. The few moist areas occurred beneath, within, or under the loose exfoliating bark of large fallen logs or between the bark and wood of upright stumps, where the three specimens were found.

*Thorius pinicola* has not been taken in sympatry with any other species of plethodontid salamander, although *Bolitoglossa macrinii* is known to occur at nearby localities.

***Remarks***: Genetic variation in *T. pinicola* was examined by Hanken (1980) (Hanken, 1983a; population 62, identity "uncertain") using protein electrophoresis. Hanken found fixed allozymic differences between *T. pinicola* and *T. longicaudus* for 4 of 18 proteins and reported a Nei genetic distance of 0.29. Similarly, he found three and four fixed

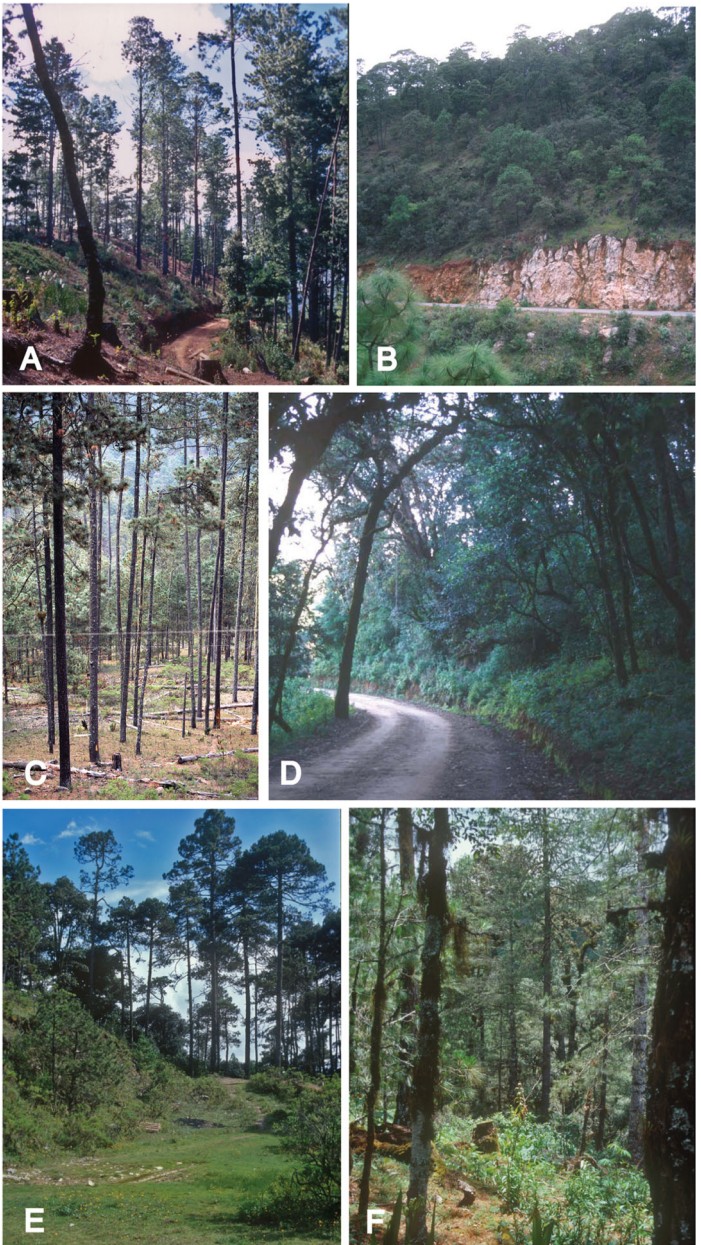

**Figure 7** **Salamander habitats in Oaxaca, Mexico.** (A) 1.7 km (by road) N of San Miguel Suchixtepec, a locality for *Thorius pinicola*, 25 January 2001. This forest is heavily logged and only three salamanders were found here this day. All were found between the bark and wood of upright stumps. The type locality is 5 km (by road) further north. (B) 19 km S of Sola de Vega, the type locality of *T. longicaudus*, 15 July 1976. Salamanders were abundant here in the 1970s, but by the 1990s the population had declined. No specimens of this species have been observed since October 1997, despite several visits to this and nearby localities. (C) 29.5 km (by road) SE of Tlaxiaco on road to San Miguel, less than 2 km from the type locality of *T. tlaxiacus*, 5 December 1978. Eighty specimens of *T. tlaxiacus* and *T. narisovalis* were collected here that day, mostly from within fallen logs. (D) 1.1 km W of Santo Tomás Teipan, the type locality of *T. minutissimus*, 23 January 2001. The previous evening, two salamanders were found in the road bank visible in the lower middle of the photograph. (E, F) Cerro San Felipe, the type locality of *T. narisovalis*, 4 August 1999. E—3 km north of La Cumbre; F—Corral de Piedra. Salamanders were found under bark on large fallen logs. (A–C) Photos by J. Hanken and (D–F) M. García-París.

differences between *T. pinicola* and two populations of *T. tlaxiacus*, its closest relative in the allozyme genetic distance-based tree; the mean genetic distance between species was 0.29. More fixed differences, and correspondingly larger genetic distances—which often exceeded 1.0—were found in comparisons with all other named taxa. With respect to the geographically closest species to the north and west, *T. narisovalis*, Hanken found fixed allozymic differences for eight of 18 proteins and a Nei genetic distance of 1.32. A complete mitochondrial genome sequence of *T. pinicola* was reported by *Mueller et al. (2004*; MVZ 231444, as *Thorius* sp. nov., GenBank accession number AY728224) and additional sequence data were reported by *Frost et al. (2006)*. *Rovito et al. (2013)* analyzed phylogenetic relationships between *T. pinicola* and congeneric species based on DNA sequence data. *Thorius pinicola* was assigned to clade 3, which presently includes 12 described and three undescribed species. It is most closely related to *T. omiltemi* and *T. grandis*, two Guerreran endemics; *T. longicaudus* and *T. tlaxiacus*, from western Oaxaca (described below); *T.* sp. 2, an undescribed species from Cerro San Felipe and San Miguel Huautla, Oaxaca; and *T.* sp. 3, an additional undescribed species from Zaachila, Oaxaca. Relationships among species in this clade, however, are not well resolved. *Thorius pinicola* is separated from topotypic *T. tlaxiacus* by a generalized time-reversible distance of 0.064 for cyt *b* and 0.027 for 16S (GTR; *Tavaré, 1986*). Comparable distances to the three other species treated below (all from their respective type localities) are larger, as follows: *T. longicaudus*, 0.073 and 0.028; *T. narisovalis*, 0.108 and 0.045; and *T. minutissimus*, 0.131 and 0.048.

The low level of mesopodial variability in *T. pinicola* (especially in the carpus, which is invariant) is exceptional for *Thorius*. Most species have moderate to high levels of carpal and tarsal variation—within species, within populations and even within individuals (right-left asymmetry; e.g., *Hanken, 1982*; *Hanken & Wake, 1998*). Carpal pattern I is the most generalized forelimb pattern observed in *Thorius* and is presumed to represent the ancestral state (*Wake & Elias, 1983*). Tarsal pattern I similarly is the state encountered in related genera and more distant outgroups (*Wake & Elias, 1983*) and is the presumed ancestral hind limb pattern for *Thorius*; it predominates in many other species of *Thorius*. Digital formulae include several instances of phalangeal loss or gain.

**Conservation status**: Based on the standard criteria used to determine the International Union for the Conservation of Nature's Red List of Threatened Species (*International Union for Conservation of Nature, 2016*), we recommend that *Thorius pinicola* be listed as Critically Endangered: there have been drastic population declines, likely exceeding 80%, at its few known localities over the last 30–40 years, which are not understood and may be continuing; the species' known Extent of Occurrence is much less than 100 km$^2$; and there is continuing decline in the extent and quality of its montane forest habitat. Further attempts to identify and assess populations of *T. pinicola* at additional localities and to more precisely define its full geographic range are urgently needed.

**Etymology**: The epithet *pinicola* is formed from the Latin words *pinus* (pine) and *-cola* (inhabitant of), in recognition of montane pine forest, which is the predominant vegetation at the type locality.

***Thorius longicaudus,*** **new species**

Long-tailed Minute Salamander

Figure 3B

*Thorius minutissimus.*—*Hanken, 1983a*: 1053.

*Thorius* sp. 4.—*Rovito et al., 2013*.

***Holotype***: MCZ A-137819, Mexico, Oaxaca, Sola de Vega District, pine-oak forest along Mexico Hwy. 131, 19 km S (by road) Sola de Vega, adult female, 16°27′35″N, 97°00′26″W, 2,200 m above sea level, 18 November 1974, J. F. Lynch, D. B. Wake and T. J. Papenfuss.

***Paratypes***: All from the type locality: MCZ A-136428, MVZ 131178, 131188, 131193, 131204, 131218, 131226, 131231, 131233, 131241, 131245, 131253, 162262, IBH 14329–30 (two specimens), same data as the holotype; MVZ 104013, 104017, 104019, 104022, 26 November 1971, J. F. Lynch; MVZ 182822–24 (three specimens), 182828, 186819–20 (two specimens), 186822–27 (six specimens), 186829–38 (10 specimens), 186843, 186849, 15 July 1976, J. F. Lynch and J. Hanken; IBH 13998, 6 October 1997, G. Parra-Olea, M. García-París and D. Wake.

***Referred specimens***: All specimens of *Thorius* from several sites near the type locality in Sola de Vega District, Oaxaca, Mexico, including the following: MVZ 104009–22 (14 specimens), Oaxaca-Puerto Escondido Rd., 11.6 mi S (by road) Sola de Vega, 16°27′52″N, 97°0′21″W, 2,100 m; MVZ 131173–257 (85 specimens), 162259–70 (12 specimens), pine-oak forest along Mexico Hwy. 131, 19 km S (by road) Sola de Vega, 16°27′35″N, 97°0′26″W, 6940 ft; MVZ 182822–54, 231675–86, Mexico Hwy. 131, 18.5 km S (by road) Sola de Vega, 16°28′00″N, 97°00′17″W, 2,150 m; and MSB 28048–51 (four specimens), 11.5 mi S (by road) Sola de Vega, 2,225 m. MVZ 182855–57 (three specimens), 183616, 183619, 15.5 km W (by road) San Vicente Lachixio, 16°45′12″N, 97°07′00″W, 2,730 m; MVZ 182859–68 (10 specimens), 13.2 km W (by road) San Vicente Lachixio, 16°45′08″N, 97°05′43″W, 2,710 m; MCZ A-148744, La Cofradía, Municipio San Pedro el Alto, 16 km beyond San Vicente Lachixio, 16°44′28″N, 97°08′32″W, 2,615 m.

***Diagnosis***: Distinguished from other species of *Thorius* by the following combination of characters: (1) large size (SL exceeds 23.5 mm in males and 24 mm in females); (2) moderately short limbs; (3) very long tail; (4) elongate, elliptical nostrils; (5) no maxillary teeth; (6) moderate number of vomerine teeth (5–10 in males and 6–10 in females); and (7) pronounced sexual dimorphism in cranial morphology.

***Comparisons***: Adult *Thorius longicaudus* are larger than *T. arboreus, T. insperatus, T. minutissimus* and *T. papaloae*. SL of adult *T. longicaudus*, and especially females, typically exceeds 25 mm, whereas most adults of the other species, and especially males, are smaller than 20 mm. The smallest known adult male *T. longicaudus*, MVZ 182823, is 22.9 mm. *Thorius adelos, T. arboreus, T. insperatus, T. macdougalli* and *T. smithi* have relatively much longer limbs (LI > 5 in *T. longicaudus*). Most *T. boreas* have relatively short tails that are the same size as or shorter than SL; TL substantially exceeds SL in all *T. longicaudus*. The nostril in *T. longicaudus* is large and elongated elliptical, whereas *T. narisovalis* has small-to-moderate-sized, round-to-oval nostrils. The nostrils are more

extremely distorted (prolate) in *T. pulmonaris* and *T. tlaxiacus*. All *T. longicaudus* lack maxillary teeth, which differentiates them from *T. adelos*, *T. aureus* and *T. smithi*, which have maxillary teeth as adults. *Thorius longicaudus* has more vomerine teeth (mean numbers in males and females are between seven and eight) than both *T. pinicola* (means between four and six) and *T. tlaxiacus* (means between four and seven).

***Description of holotype***: Head width 3.3 mm; snout to gular fold (head length) 4.5 mm; head depth at posterior angle of jaw 2.2 mm; eyelid width 0.9 mm; eyelid length 1.7 mm; anterior rim of orbit to snout 1.2 mm; horizontal orbit diameter 1.2 mm; interorbital distance 1.2 mm; distance between corners of eyes 1.8 mm; distance separating external nares 1.0 mm; major axis of nostril 0.7 mm; minor axis of nostril 0.3 mm; snout projection beyond mandible 0.6 mm; snout to posterior angle of vent (SL) 27.7 mm; snout to anterior angle of vent 25.1 mm; snout to forelimb 7.8 mm; axilla to groin 14.7 mm; LI 6 costal interspaces; shoulder width 2.9 mm; TL 39.2 mm; tail width at base 3.0 mm; tail depth at base 2.6 mm; forelimb length (to tip of longest toe) 4.1 mm; hind limb length 4.9 mm; hand width 0.9 mm; foot width 1.2 mm. Numbers of teeth: premaxillary 3; maxillary 0; vomerine 4-5.

Ground color of head, body and tail blackish-brown (Fig. 3B). Paler brown dorsal stripe with indistinct borders begins on nape and extends posteriorly, more obscure towards tip of tail. Venter pale brown; scattered white spots extend dorsally onto sides of head, trunk and tail. Limbs dark brown dorsally, slightly paler ventrally. Costal grooves, gular fold and extension of fold onto neck without pigment; otherwise, without distinguishing marks. Parotoid gland distinct but not differentially colored.

***Variation***: Mean adult SL 25.0 mm (range 23.6–28.3) in 10 males, 25.5 mm (24.4–27.7) in 10 females. Head relatively narrow; SL 8.1 times head width (6.9–8.8) in males, 8.3 times head width (8.1–8.6) in females. Snouts pointed to bluntly pointed. Nostrils relatively large, elliptical; ratio of major to minor axes 1.8 (1.5–2.0) in males and 1.8 (1.4–2.3) in females. Eyes moderately large, protrude slightly beyond jaw margin in dorsal view. Suborbital groove intersects lip on each side of head. Premaxillary teeth 1.1 (1–2) in adult males, 1.8 (0–4) in females. No maxillary teeth. Vomerine teeth 7.3 (5–10) in males, 7.9 (6–10) in females. Limbs moderately long; LI 5.3 (5.0–5.5) in males, 5.5 (5.0–6.0) in females. Manus and pes relatively well developed; foot width 1.2 mm in both males (1.1–1.2) and females (1.0–1.3). Digits 1 and 4 (manus) and 1 and 5 (pes) short and fused to neighboring digit; central digits long and separate from one another, with rounded tips. Digits on manus, in order of decreasing length, 3-2-4-1; toes 3-4-2-5-1. Tail long—greatly exceeds SL—and tapered; SL divided by TL 0.69 (0.63–0.73) in 10 males, 0.76 (0.62–0.91) in 10 females. Mental gland indistinct in adult males. Postiliac gland small, pale, inconspicuous. Parotoid glands indistinct to very evident in many specimens, including the holotype.

***Coloration in life***: A distinct, tan-reddish stripe with coppery-brassy highlights and indistinct dark chevrons extends anteriorly from back of head; head stripe with fine tan border sharply demarcated dorsolaterally from unmarked black upper flanks; whitish flecks lower on flanks; densely packed whitish markings form a wash laterally, which

continues less densely onto venter. Iris reddish brown. Slight reddish brown pigment at limb insertions (D. Wake field notes, 5 October 1997; IBH 13998, gravid female, 22 mm SL with partially regenerated tail). Ventral coloration dark with whitish flecks (J. Hanken field notes; 15 July 1976; MVZ 182822–24, 182828, 186819–20, 186822–27, 182829–38, 182843 and 182849; Fig. 4B).

*Coloration in preservative*: A moderately dark species, with a distinct, paler dorsal band extending from the otic region to the tip of the tail. The band is interrupted by obscure herringbone markings in some individuals, and often there is a thin, median dark line. The venter is paler than the flanks and contains numerous white spots on the belly or gular region in most individuals. A pale nuchal spot is present in most individuals.

*Osteology*: There is considerable sexual dimorphism in cranial morphology. The skull is poorly ossified, especially in males (Figs. 5D–5F). Ascending processes of the single premaxillary bone remain separate in 8 of 10 females but in only 3 of 10 males (character 1, state a); they articulate or fuse to varying degrees in remaining specimens (states b–d). Dental processes of the premaxilla are separate from the maxilla in all males but in only two females (character 2, state a); the bones overlap in ventral view or articulate in most females (states b–d). The premaxilla bears teeth in all specimens (character 8, state b). In both sexes, the nasal bone is thin and rod-like (character 3, state b) or slightly broader and extending somewhat anteriorly over the nasal capsule (state c). Nasal and maxilla are separate in nearly all males but in fewer than half the females (character 4, state a); the bones barely articulate (state b) in all remaining specimens except one female, in which they are fused (state c). The prefrontal is divided on one or both sides of several specimens (both sexes), and remains separate from the nasal in nearly all males and in slightly more than half the females (character 5, state b). It articulates with the nasal (state c) in all remaining specimens except one female, in which the bones are fused (state d). The prefrontal is well separated from the maxilla (character 6, state a) in nearly all males but in only half the females; the bones articulate in all remaining specimens (state b). The septomaxillary bone is barely visible on one side of one male (character 7, state b) and is lacking in all other specimens (state a).

Maxillary bones are delicate—long and slender—and taper to a point posteriorly. There are no maxillary teeth (character 9, state a). The vomer is reasonably well developed. The preorbital process of the vomer, when present, is short and bears teeth. There are very few vomerine teeth, which are arranged in a short row diagonally toward the midline. The frontal fontanelle is relatively narrow. The parietal fontanelle is very broad in males (mean breadth 0.55 times maximum skull width across parietals; range 0.48–0.68) but slightly narrower in females (mean 0.47, range 0.43–0.66). There is no crest on the occipito-otic and no columellar process on the operculum. Postsquamosal process is well developed.

There are fourteen presacral vertebrae. Typically, all trunk vertebrae but the last bear ribs; in a few specimens, the last trunk vertebra has a partial rib on one or both sides. Mean number of caudal vertebrae 35.0 (range 33–37) in five males, 36.8 (31–45) in five females (Fig. 8).

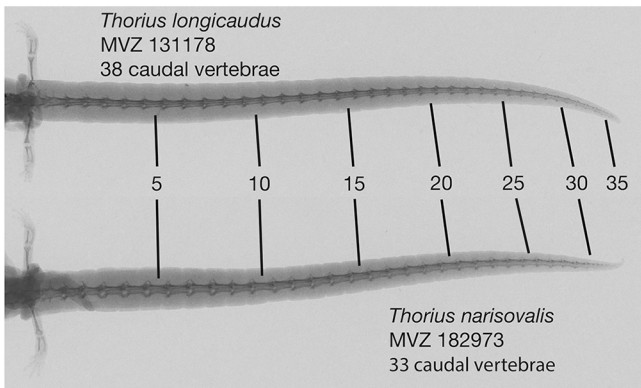

**Figure 8** **Radiographs of tails of two adult *Thorius*.** *Thorius longicaudus* has a longer tail, relative to standard length, and more caudal vertebrae (mean 36.8, range 31–45) than *T. narisovalis* (33.0, 31–36). Both specimens are adult females of similar body size: MVZ 131178—SL 25.1, tail length (TL) 35.5; MVZ 182973—SL 29.0, TL 37.8.

Limbs are slender but well developed. A tibial spur is present as an attached crest in most specimens, but it ranges from well developed to absent in few others.

Mesopodial morphology is moderately variable, with several variant patterns in both the wrist and ankle. The predominant carpal pattern is I (77%; Fig. 6A). Three other patterns, each with one additional fusion relative to pattern I, occur at moderate to low frequencies: II (fused distal carpals 1-2 and 3; 15%), III (fused distal carpals 3 and 4 plus centrale; 5%) and V (fused distal carpals 1-2, 3, and 4 plus centrale; 3%). Modal tarsal pattern I (85%; Fig. 6B). Four other patterns, each with one additional fusion relative to pattern I, are found in only one or two tarsi each: II (centrale fused to distal tarsal 4-5; 3%), III (fused distal tarsals 1-2 and 3; 3%), IV (fused intermedium plus centrale; 3%) and V (fused intermedium plus fibulare; 5%; Fig. 6C).

The digital skeleton is invariant in both forelimbs and hind limbs. Phalangeal formulae are 1-2-3-2 (manus) and 1-2-3-3-2 (pes). Limb bone epiphyses and mesopodial elements are mineralized in most adults.

***Distribution and ecology***: *Thorius longicaudus* is known from two geographic areas; both are in the state of Oaxaca. The first is the vicinity of the type locality, which is along Mexico Hwy. 131 approximately 19 km south of the village of Sola de Vega, in one of the northernmost ridges of the Sierra Madre del Sur of southeastern Mexico (Figs. 1 and 7B). A second area is near San Vicente Lachixio, about 40 km to the northwest. At the type locality, salamanders have been collected along a dirt road heading east from the main highway near the top of the ridge, opposite a microwave station. Recorded elevations range approximately from 2,100 to 2,200 m (elevation at the type locality was recorded initially as 2,200 m, but it has been recorded subsequently as low as 2,085 m). The dominant natural habitat is pine-oak forest, although much of the vegetation has been extensively cleared; only scattered trees remain.

*Thorius longicaudus* was at one time very abundant at the type locality and adjacent forests; large series, deposited at MVZ, LACM and MSB, were collected in the 1970s. Recent visits document a precipitous population decline, and the species is now virtually

impossible to find (*Parra-Olea, García-París & Wake, 1999*; *T. minutissimus*). In November 1974, the open stand of forest at the type locality was very dry and the road dusty. Roadside banks were dry except in deep shade. Fewer than 10 *Thorius* were found under bark of logs and under log chips, and none was found under rocks. Salamanders were abundant, however, under crusted dirt on the road bank; approximately 300 individuals were seen, mostly clustered together in bunches of 10–12 each in deep cracks, sometimes with three or four salamanders in direct contact with one another. In July 1976, 75 specimens were found in approximately two hours. Salamanders were abundant in a stand of pines under or inside logs (including under bark), under small fallen branches and under rocks or small piles of cow dung exposed to the sun. Several were found in the small moist area where two fallen branches overlapped one another or where a small branch contacted the ground, but none was found in the nearby road bank. By October 1997, the habitat had changed considerably following extensive logging and widening of the road. We searched for more than two hours, turning abundant cover of wood chips, chip piles, logs and bark on logs, as well as leaf litter and rocks. Conditions seemed good with adequate moisture but only a single individual was found, under a small pine log lying beside the road. We found no salamanders during any of our subsequent visits (four since 2000, at different times of the year, including June 2014), either in disturbed habitats or in the remaining fragments of natural (original) habitat.

Sherman Minton and Charles Bogert first collected *Thorius* near San Vicente Lachixio in 1966 (CM 68248–51), and salamanders were abundant when James Lynch and James Hanken visited the area in the 1970s (large series in MVZ). Populations have declined precipitously since then. One of us (S.R.) spent an afternoon searching in the San Vicente Lachixio area in June 2014 but failed to find even a single salamander of any kind. The forest appeared to be intact, with abundant forest cover, leaf litter, and fallen logs. *Thorius longicaudus* was last seen alive in the field in March 1998 (MCZ A-148744).

*Thorius longicaudus* is the only species of its genus known from the type locality, where two other plethodontid species, *Pseudoeurycea conanti* and *P. cochranae*, have been taken in sympatry (*Parra-Olea, García-París & Wake, 1999*). *Bolitoglossa oaxacensis* is found a few kilometers to the north and to the southwest (*Parra-Olea, García-París & Wake, 2002*). At San Vicente Lachixio, *T. longicaudus* is sympatric with *T. tlaxiacus* (described below), and both *P. cochranae* and *P. anitae* occur in the same area.

***Remarks***: Genetic variation in *T. longicaudus* was examined using protein electrophoresis (*Hanken, 1980*; *Hanken, 1983a*: population 64, 20 individuals, listed as *T. minutissimus*; populations 65 and 66, 10 and five individuals, respectively, listed as "uncertain"). The two population segments of this species differ by a Nei D of 0.21. Hanken found fixed allozymic differences for 4 of 18 proteins between *T. longicaudus* and *T. pinicola* from their respective type localities (sample size also 20 individuals for *T. pinicola*). The Nei genetic distance to *T. pinicola* equals 0.29. The species differs from *T. tlaxiacus* in having 3 of 18 proteins with fixed differences and a Nei D of 0.34–0.53 for the different populations. Even larger differences were found to all other named taxa. For example, the geographically proximate *T. narisovalis* (which occurs well to the north and west;

five samples of 16–20 individuals each) differs from both new species by fixed differences for 9 of the 18 proteins studied and Nei distances greater than 1.0. *Rovito et al. (2013)* analyzed phylogenetic relationships between *T. longicaudus* and congeneric species based on analysis of DNA sequence data. *Thorius longicaudus*, a member of clade 3, is most closely related to six other species endemic to southern and western Oaxaca and Guerrero: *T. pinicola*, *T. grandis*, *T. omiltemi*, *T. tlaxiacus* (described below), and two undescribed species, *T.* sp. 2 and *T.* sp. 3. Relationships among these species, however, are not well resolved. *Thorius longicaudus* is separated from topotypic *T. tlaxiacus* by a GTR distance of 0.045 for cyt *b* and 0.031 for 16S. All but one distances to the three other species treated in this paper (all from their respective type localities) are larger, as follows: *T. pinicola*, 0.073 and 0.028; *T. narisovalis*, 0.116 and 0.038; and *T. minutissimus*, 0.142 and 0.039.

Evolutionary consequences of miniaturization of adult body size for cranial and appendicular morphology were examined by *Hanken (1982)* (*Hanken, 1983b*; *Hanken, 1984*; *Hanken, 1985*; *T. minutissimus*). Absence of variation in digital skeletal formulae in both forelimbs and hind limbs in *T. longicaudus* is unusual for *Thorius*; most species exhibit at least moderate variation among and within individuals (*Hanken, 1982*; *Hanken & Wake, 1998*).

*Conservation status*: Based on the standard criteria used to determine the IUCN Red List of Threatened Species (*International Union for Conservation of Nature, 2016*), we recommend that *Thorius longicaudus* be listed as Critically Endangered: there have been drastic population declines, likely exceeding 80%, at its few known localities over the last 30–40 years, which are not understood and may be continuing; and there is continuing decline in the extent and quality of its montane forest habitat. The species is known from two circumscribed geographic areas, and while these areas are about 40 km apart, the full Extent of Occurrence is unknown at this time and the Area of Occupancy may nevertheless be very small. Further attempts to identify and assess populations of *T. longicaudus* at additional localities and to more precisely define its full geographic range are urgently needed.

*Etymology*: The epithet "longicaudus," is derived from the Latin "longus" ("long") and "cauda" ("tail"), and refers to the long tail that is a conspicuous feature of these salamanders as adults.

### *Thorius tlaxiacus*, new species
Heroic Minute Salamander
Figure 3C
*T.* sp. 5.—*Rovito et al., 2013*.

*Holotype*: MCZ 148746, Mexico, Oaxaca, Tlaxiaco District, 27.3 km SSE (by road) Tlaxiaco, adult female, 17°8′54″N, 97°37′12″W, 2,855 m above sea level, 22 July 1999, G. Parra-Olea, M. García-París, D. B. Wake and J. Hanken.

*Paratypes*: All from Oaxaca, Mexico: MVZ 183443–51 (nine specimens), 29.5 km SE (by road) Heroica Ciudad de Tlaxiaco on road to San Miguel, 17°6′5″N, 97°36′55″W, 3,080 m

above sea level, 5 December 1978, James Hanken and Thomas Hetherington; MCZ A-148745, same data as the holotype.

***Referred specimens***: All from Oaxaca, Mexico: MVZ 183614–15 (two specimens), 183617–18 (two specimens), 183620–23 (four specimens), 185319, 187108–16 (nine specimens), 15.5 km W (by road) San Vicente Lachixio, Sola de Vega District, 16°45′12″N, 97°7′0″W, 2,730 m, 16 July 1976, J. F. Lynch and J. Hanken; MCZ A-148747–48 (two specimens), 13 km W San Vicente Lachixio, 16°44′52″N, 97°5′18″W, 2,720 m, 26 January 2001, G. Parra-Olea, M. García-París, J. Hanken and T. Hsieh.

***Diagnosis***: Distinguished from other species of *Thorius* by the following combination of characters: (1) very large size (SL averages nearly 28 mm in both males and females); (2) limbs moderately short; (3) moderately long tail (TL slightly exceeds SL in most adults); (4) prolate nostrils (mean ratio of major to minor axes exceeds 2.0 in both males and females; (5) no maxillary teeth; and (6) moderate number of vomerine teeth (4–6 in males, 4–8 in females).

***Comparisons***: Adult *Thorius tlaxiacus* are among the largest species of *Thorius* and they have the most extremely distorted nostrils (prolate, shared only with *T. pulmonaris*). Their moderately short limbs (LI 5–6) differentiate them from the shorter limbed *T. aureus*, *T. boreas* and *T. minutissimus*, and from the longer limbed *T. adelos*, *T. arboreus*, *T. insperatus*, *T. macdougalli*, *T. papaloae* and *T. smithi*. Based on external morphology, it is difficult to differentiate *T. tlaxiacus* from the other, somewhat smaller species named herein, *T. longicaudus* and *T. pinicola*, although the latter two species have elongated elliptical rather than prolate nostrils, and *T. longicaudus* has a slightly longer tail. All three new species, however, are well differentiated genetically from one another (see Remarks section of each species account).

***Description of holotype***: Head width 3.6 mm; snout to gular fold (head length) 4.9 mm; head depth at posterior angle of jaw 2.3 mm; eyelid width 0.6 mm; eyelid length 1.4 mm; anterior rim of orbit to snout 1.1 mm; horizontal orbit diameter 0.9 mm; interorbital distance 1.4 mm; distance between corners of eyes 2.0 mm; distance separating external nares 0.8 mm; major axis of nostril 0.6 mm; minor axis of nostril 0.3 mm; snout projection beyond mandible 0.3 mm; snout to posterior angle of vent (SL) 29.0 mm; snout to anterior angle of vent 27.9 mm; snout to forelimb 7.4 mm; axilla to groin 16.9 mm; LI 6; shoulder width 2.7 mm; TL 21.0 mm (tip missing); tail width at base 2.9 mm; tail depth at base 2.8 mm; forelimb length (to tip of longest toe) 4.5 mm; hind limb length 4.8 mm; hand width 0.9 mm; foot width 1.5 mm. Numbers of teeth: premaxillary 0; maxillary 0; vomerine 3-3.

Ground color of head, body and tail dark grey-brown (Fig. 3C). Prominent golden-brown dorsal stripe arises from a golden spot on the nape, extends to the middle of the tail where it becomes diffuse. Venter is grey, lighter than the flanks, with obscure light-grey mottling in the gular area. Venter pale brown; scattered white spots extend dorsally onto sides of head, trunk and tail. Limbs are about the same color as the flanks, but a little lighter. Prominent y-shaped mark arises from the eyes, fuses at the back of the head and makes a short middorsal stripe to the golden spot on the nape. Parotoid gland is

prominent, but paler than the surrounding tissue. Snout has a bright, V-shaped patch of color at its very tip; arms of the V point to the eyes. Light white speckling between the eyes dorsally. Nasolabial protuberances are unpigmented.

*Variation*: Mean adult SL 28.0 mm (range 21.1–30.2) in seven males, 27.7 mm (22.6–31.0) in four females. Head narrow; SL 8.3 times head width (7.5–9.2) in males, 8.0 times head width (7.3–8.6) in females. Snouts pointed to bluntly pointed. Nostrils large, prolate; ratio of major to minor axes 2.1 (1.7–2.5) in males and 2.3 (2.0–2.5) in females. Eyes moderately sized, do not protrude beyond jaw margin in dorsal view. Suborbital groove intersects lip on each side of head. Premaxillary teeth 1.3 (0–2) in adult males, 0.5 (0–1) in females. No maxillary teeth. Vomerine teeth 4.9 (4–6) in males, 6.3 (4–8) in females. Limbs stout and moderately short; LI 5.3 costal interspaces (5.0–5.5) in males, 5.5 (5.0–6.0) in females. Manus and pes well developed; foot width 1.3 mm in males (1.0–1.6) and 1.4 in females (1.2–1.5). Digit 1 is well developed, especially on pes; outermost digit of both manus (4) and pes (5) is discrete but very short and fused to neighboring digit. Interior digits have rounded free tips with distinct subterminal pads. Digits on manus, in order of decreasing length, 3-2-4-1; toes 3-4-2-1-5. Tail moderately long and tapered; SL divided by TL 0.95 (0.84–1.04) in five males, 0.96 (0.87–1.05) in two females. Mental gland large and distinct in adult males. Postiliac gland obscure. Prominent parotoid glands form elongate, lightly pigmented swellings at the posterolateral margin of the head.

*Coloration in life*: Based on field notes of D. B. Wake, 26 July 1999. MCZ 148746: Very dark brown ground color. Broad dark-brown stripe starts in nuchal region with a chestnut-colored spot. Paratoid glands have small chestnut-colored streak. Extensive fine white spotting and streaking laterally, especially around forelimb insertions and on neck. A few fine white spots ventrally on very dark belly and on lighter, grey-black throat; more numerous on tail venter. Iris black. MCZ A-148745: Ground color black with distinctly reddish-brown, relatively bright dorsal stripe. Border between stripe and lateral surfaces sharp but scalloped. Stripe interrupted in places along midline by black ground color, which makes chevron-like marks. Fine white speckling laterally but less dense on belly. Limb insertions reddish. Paratoid glands large, separated by dark, lyrelike pattern. Iris dark.

*Coloration in preservative*: MCZ A-148745: much like holotype, but somewhat lighter in general coloration. Bright golden spot on nape, which gives rise to dorsal stripe marked by dark chevrons at midline. V-shaped light patch on snout present but less conspicuous than in holotype. Most of the remainder of the type series is damaged (tissue taken for genetic analysis), so coloration is hard to describe. Many specimens have golden spot on nape and V-shaped mark on snout.

*Osteology*: Based primarily on a μCT scan of MVZ 183447, an adult male whose tail, viscera and ventral body wall were removed earlier for genetic analysis. Only the anterior portion of the body was scanned. Hence, data are unavailable for most of the postcranial skeleton. Vertebral counts were made from digital radiographs of MCZ A-148745 and A-148746, both adult females. The skull is poorly ossified; several bones, especially

rostrally and dorsally, are thin and delicate and fail to articulate with one another (Figs. 5G–5I). Ascending processes of the single premaxillary bone are fused along less than one-half of their length; the internasal fontanelle is moderately sized (character 1, state b). Dental parts of the premaxilla are well separated from the maxilla (character 2, state a). The premaxilla bears two mature (ankylosed) teeth rostral to a third, unerupted (successional) tooth (character 8, state b). Nasal bones extend somewhat anteriorly from the posterior edge of the cartilaginous nasal capsule (character 3, state b). Both are irregularly shaped, each consisting of a broad but thin dorsal part with an uneven anterior border, and a stouter ventral part; the two parts are connected by a thin bridge. Nasal and maxilla are barely separated on the left side (character 4, state a) but articulate slightly on the right (state b). Each prefrontal is a thin, crescent-shaped bone that is separate from both the nasal (character 5, state b) and the maxilla (character 6, state a). The septomaxillary bone is absent on both sides (character 7, state a).

Maxillary bones are very slender; they appear scimitar-shaped in lateral view (i.e., curved rather than straight). There are no maxillary teeth (character 9, state a). The vomer is especially slender and bears very few teeth, which are arranged in a short row diagonally toward the midline. The frontal fontanelle is relatively narrow but the parietal fontanelle is very wide; breadth is 0.50 times maximum skull width across parietals). There is no crest on the occipito-otic and no columellar process on the operculum. In comparison to most other congeners, the lower jaw is stout whereas the postsquamosal process is small.

Forelimbs are slender but well developed; long bone epiphyses and carpal elements are mineralized. Phalangeal formulae (manus) is 1-2-3-2 on both sides. Neither carpal pattern can be scored reliably. There are 14 presacral and two caudosacral vertebrae; the tail tip was removed at caudal 18 (MCZ A-148745) or 17 (MCZ A-148746).

***Distribution and ecology***: *Thorius tlaxiacus* is known from two geographic areas: the type locality and adjacent localities near Heroica Ciudad de Tlaxiaco, in west-central Oaxaca; and about 80 km to the southeast, near the village of San Vicente Lachixio, Oaxaca (Fig. 1). Recorded elevations range from 2,665 to 3,080 m (Tlaxiaco) and from 2,720 to 2,730 m (San Vicente Lachixio). According to field notes of J. Hanken, 6 December 1978, *Thorius* were abundant the previous day in pine-oak forest at 29.5 km SE of Tlaxiaco. The best collecting spots were partially disturbed slopes with abundant fallen logs (Fig. 7C). Salamanders were most abundant inside fragmenting fallen logs; a few others were found under bark. As many as 10 or 12 specimens were taken together from a single crevice. We revisited the Tlaxiaco area in July 1999 and found that much of the naturally occurring pine-oak forest had been extensively cleared, leaving only scattered trees. Only three specimens of *T. tlaxiacus*, including the holotype, were seen. One of us (S.R.) visited Tlaxiaco and San Vicente Lachixio localities in July 2014 but saw no salamanders in either area.

*Thorius tlaxiacus* is sympatric with a second terrestrial species of *Thorius* at each area: *T. narisovalis* (Tlaxiaco) and *T. longicaudus* (San Vicente Lachixio). Additional plethodontid species sympatric at San Vicente Lachixio are *Pseudoeurycea cochranae* and *P. anitae.*

***Remarks:*** Genetic variation in *T. tlaxiacus* was examined using protein electrophoresis (*Hanken, 1980*; *Hanken, 1983a*; populations 61 and 63, comprising nine and 18 individuals, respectively, listed as "uncertain"). The Nei genetic distance between the two populations of *T. tlaxiacus* was 0.15, largely reflecting a fixed allozymic difference at one protein. Hanken found fixed allozymic differences for 3 or 4 of 18 proteins between *T. tlaxiacus* and topotypic *T. pinicola* (population 62, sample size 20 individuals), its closest relative in the allozyme genetic distance-based tree. The corresponding average genetic distance was 0.29; larger genetic differences, often exceeding 1.0, were found to all other species. For example, the average pairwise distance to *T. longicaudus*, the next most similar species to *T. tlaxiacus*, equals 0.45. *Rovito et al. (2013)* analyzed phylogenetic relationships between *T. tlaxiacus* and congeneric species based on analysis of DNA sequence data. *Thorius tlaxiacus* clustered with six other species within clade 3; all are endemic to Oaxaca or Guerrero: *T. grandis*, *T. omiltemi*, *T. longicaudus*, *T. pinicola* and two undescribed species, *T.* sp. 2 and *T.* sp. 3. Relationships among these species, however, are not well resolved. *Thorius tlaxiacus* is separated from topotypic *T. longicaudus* by a GTR distance of 0.045 for cyt *b* and 0.031 for 16S. All but one distances to the three other species treated in this paper (all from their respective type localities) are larger, as follows: *T. pinicola*, 0.064 and 0.027; *T. narisovalis*, 0.118 and 0.043; and *T. minutissimus*, 0.134 and 0.044.

One of two sympatric species of *Thorius* from localities west of San Vicente Lachixio is assigned to *T. tlaxiacus* based on allozyme and DNA sequence data (*Hanken, 1983a*; *Rovito et al., 2013*). The same data have been used to confirm the species identification of all referred specimens from these localities as well as the entire type series from Tlaxiaco (see above). In addition, we provisionally assign the following specimens to *T. tlaxiacus* based solely on external morphology: MVZ 182990, 182993–94 (two specimens), 182997, 183000–02 (three specimens), 183005, 183010, 185368 and 185371, 29.5 km SE (by road) Heroica Ciudad de Tlaxiaco on road to San Miguel el Grande.

***Conservation status:*** Based on the standard criteria used to determine the IUCN Red List of Threatened Species (*International Union for Conservation of Nature, 2016*), we recommend that *Thorius tlaxiacus* be listed as Critically Endangered: there have been drastic population declines, likely exceeding 80%, at its few known localities over the last 30–40 years, which are not understood and may be continuing; and there is continuing decline in the extent and quality of its montane forest habitat. The species is known from two circumscribed geographic areas, and while these areas are about 80 km apart, the full Extent of Occurrence is unknown at this time and the Area of Occupancy may nevertheless be very small. Further attempts to identify and assess populations of *T. tlaxiacus* at additional localities and to more precisely define its full geographic range are urgently needed.

***Etymology:*** The name of the species is derived from the name of the city nearest to the type locality, Heroica Ciudad de Tlaxiaco, an important regional center in colonial Mexico.

# REDESCRIPTIONS

Original descriptions of *Thorius minutissimus* and *T. narisovalis*, two species endemic to central and southern Oaxaca, were brief and limited to relatively few external characters. They are difficult to apply reliably for identification of most populations from beyond the respective type localities, and these names have been applied frequently and erroneously to populations that belong to neither species. Further study of these and other named species, formal description of several new species from Oaxaca, and the availability of recently collected specimens now enable more accurate characterization of both species. We provide a redescription of each species in order to facilitate accurate identification of these and other species.

### *Thorius minutissimus* Taylor, 1949

Extremely Minute Salamander

Figure 3D

*Holotype*: AMNH 52673, Mexico, Oaxaca, "Santo Tomás Tecpan," adult female, 3 March 1946, T. C. MacDougall.

*Additional specimens examined*: IBH 23011–12 (two specimens), Mexico, Oaxaca, 1.1 km W (by road) Santo Tomás Teipan, 16°09′1.8″N, 95°35′34.4″W, 2,458 m.

*Diagnosis*: Distinguished from other species of *Thorius* by the following combination of characters: (1) moderate size (SL less than 24 mm in females; the only known adult male is 19 mm); (2) short limbs, with a very short outside digit in the manus (digit 4) and pes (digit 5); (3) short tail; (4) oval nostrils; (5) no maxillary teeth; (6) moderate number of vomerine teeth; and (7) reddish dorsal stripe.

*Comparisons*: Adult *T. minutissimus* are smaller than *T. pinicola*, *T. boreas*, *T. narisovalis* and *T. longicaudus*. The largest-known adult *T. minutissimus*, a female, is 23.6 mm SL. The other species typically exceed 24 mm and most females exceed 25 mm. *Thorius adelos*, *T. arboreus*, *T. insperatus*, *T. macdougalli*, *T. magnipes* and *T. smithi* have much longer limbs; LI is 4.5 or less. LI is 5.5 or more in *T. minutissimus*. *Thorius longicaudus*, *T. magnipes*, *T. narisovalis*, *T. papaloae*, *T. pennatulus* and *T. pinicola* have relatively long tails that exceed SL in nearly all adults. TL in *T. minutissimus* typically is the same as or shorter than SL. The nostril in *T. minutissimus* is large and oval, whereas *T. narisovalis* has small-to-moderate-sized, round-to-oval nostrils. All *T. minutissimus* lack maxillary teeth, which differentiates them from *T. adelos*, *T. aureus*, *T. schmidti* and *T. smithi*, which have maxillary teeth as adults. In life, *T. minutissimus* has a reddish dorsal stripe, which distinguishes this species from *T. dubitus*, which has a greenish stripe. The species is smaller than *T. pulmonaris* and *T. tlaxiacus* and has oval rather than prolate nostrils.

*Variation*: Mean adult female SL 23.0 mm (range 22.3–23.6). Head moderately broad; SL 8.0 times head width (7.9–8.0). Snouts rounded to roundly pointed. Nostrils relatively large and oval, but not elliptical; ratio of major to minor axes 1.29 (1.2–1.4). Eyes moderately large, protrude slightly beyond jaw margins in dorsal view. Suborbital groove intersects lip on each side of head. One premaxillary tooth in each female, no maxillary

teeth, and seven (5–9) vomerine teeth. Limbs moderately long; LI 6.0 costal interspaces (5.5–6.5). Manus and pes narrow; pes width 0.95 mm (0.9–1.0). Digits 1 and 4 (manus) short. Digit 1 (pes) short, tip not free of webbing; digit 5 reduced to slight bulge at base of digit 4. Central digits (all limbs) relatively long with rounded tips. Fingers, in order of decreasing length, 3-2-4-1; toes 3-2-4-1-5. Tail moderately long and tapered; SL divided by TL 1.03 (0.94–1.12). Postiliac gland small, pale and relatively inconspicuous. Parotoid glands prominent in some specimens, but less so in others.

*Coloration in life*: Ground color of flanks very dark blackish brown. Dark brown dorsal stripe from snout to tip of tail, widest in head region, narrow over shoulders. Dark, regular, herringbone pattern interrupts dorsal stripe, especially over trunk. Obscure reddish nuchal spot. Venter slightly paler than flanks, with fine pale stippling; superficially, appears unspotted. Overall color pattern dark brown (J. Hanken field notes, 23 January 2001; IBH 23011–12; Fig. 4C).

*Coloration in preservative*: Based on IBH 23011–12, both adult females (Fig. 3C). Dorsal ground color dark blackish grey. Obscure dorsal stripe—only slightly paler than ground color—from nape to base of tail. Venter dark grey with numerous pale spots, especially in gular region. Indistinct nuchal spot. IBH 23011 with pair of small pale areas over shoulders.

*Osteology*: Based on IBH 23011 (µCT scan) and 23012 (cleared and stained), both adult females. Skull weakly ossified, except for well-developed otic capsules (Figs. 5J–5L). Ascending processes of premaxillary bone arise from the dental process by only one root. They are fused and twisted for much of their length (character 1, state d) and bear a small slit-like fontanelle mid-length. Dental process of premaxilla separate from maxilla in ventral view in IBH 23012 (character 2, state a), but these elements overlap in ventral view and articulate in IBH 23011 (state d). Premaxillary teeth absent in IBH 23012 (character 8, state a), but IBH 23011 has one mature (ankylosed) tooth on the right side and two unerupted (successional) teeth on the left (state b). Nasal and maxillary bones separate (character 4, state a). Prefrontal bone separate from nasal (character 5, state b) except on the left side of IBH 23011, where the two bones barely articulate (state c). Prefrontal separate from maxilla (character 6, state a). Septomaxillary bone absent (character 7, state a) except barely visible on the right side of IBH 23011 (state b). Maxillary bone long, lacks teeth (character 9, state a). Vomer well developed except for rudimentary preorbital process. Vomerine teeth 2–4 per side, arranged in slightly curved row at caudal end of bone. Facial parts of frontal well developed; dorsal parts extremely thin with highly uneven medial margin. Frontal fontanelle moderately wide. Parietal fontanelle wide; breadth 0.6 times maximum skull width across parietals. Rudimentary columellar process on each operculum. Occipito-otic without crest, postsquamosal process well developed. Hyobranchial cartilages are not mineralized.

Fourteen trunk vertebrae, all with ribs. Each rib on last trunk vertebra has only one head. Two caudosacral vertebrae; tail tip was removed at caudal 14 (IBH 23012) or 17 (IBH 23011).

Limbs slender but well developed. Tibial spur well developed.

Mesopodial morphology is identical in the two forelimbs (carpal pattern I) but differs between the two hind limbs (Figs. 6C and 6D). The left side has tarsal pattern V (see above, *T. pinicola*), whereas the right side has pattern VII, with one additional fusion relative to pattern V (fused distal tarsal 4-5 and centrale). A distinct crease remains visible between intermedium and fibulare.

Digital phalangeal formulae 1-2-3-2 (manus) and 1-2-3-2-1 (pes). Penultimate phalange on third toe of each pes very short. Limb bone epiphyses are mineralized. Mesopodial cartilages not mineralized.

***Distribution and ecology*:** *Thorius minutissimus* is known only from the immediate vicinity of the type locality (Figs. 1 and 7D; *Lamoreux, McKnight & Cabrera Hernandez, 2015*). The following account is based on field notes by J. Hanken from 25 January 2001. Two specimens (IBH 23011–12) were collected in a patch of oak forest where the dirt road crosses a ridge above the village of Santo Tomás Teipan. The habitat consists of many tall, mature trees, a dense understory of tall bushes and ferns, many bromeliads high up on the trees, and abundant logs and trunks on the forest floor. Both specimens were taken in the early evening (19:00–20:00 h) from a 0.5–1 m high road bank on the south side of the road. The first salamander was captured after it partially emerged from small hole in the road bank. The second specimen was obtained by randomly digging into the bank at several places. No salamander was found in the adjacent forest during the day. In addition, a single individual was found during a conservation assessment in April 2009 (*Lamoreux, McKnight & Cabrera Hernandez, 2015*). *Thorius minutissimus* has not been found sympatric with any other plethodontid species, although *Bolitoglossa zapoteca* is known from near the type locality.

***Remarks*:** We have examined the holotype and one paratype (AMNH 52673–74), as well as a third specimen collected in 1955 that bears a UIMNH tag (37370) but is now housed at MCZ (A-30869), and the two recently collected specimens described above. The type series was poorly preserved (as noted by *Taylor, 1949*) and the holotype has a mutilated mouth and lacks all or most of each limb. The above paratype also lacks limbs. Another paratype (formerly AMNH A53930) is now at KU (28080). Four additional specimens were taken at or near the type locality: FMNH 105258–61, 105636, Santo Tomas Teipan, "2 leagues E of Tlahuilotepec," collected by T. C. Macdougall in 1942. These specimens also are poorly preserved and badly damaged, but we assign them to *T. minutissimus* because only one species of *Thorius* is known from this region and these specimens appear to resemble the type series of *T. minutissimus.*

The history of the name *minutissimus* is complicated. Taylor apparently intended to name *T. minutissimus* after T. C. MacDougall, who collected the original type series in 1946 (1949: 1). Instead, he honored MacDougall by naming an even smaller species of *Thorius* from northern Oaxaca after him, *T. macdougalli*. Both new species were described in the same publication (*Taylor, 1949*). In the introductory section, Taylor indicates that *T. minutissimus* is from Cerro Humo, a mountain in the Sierra de Juárez, northern Oaxaca. In the body of the paper, however, he describes populations from Cerro Humo as *T. macdougalli*, a small species found only in the northern part of the state, and identifies

the type locality of *T. minutissimus* as being from the extreme southeastern extent of the range of the genus. No additional topotypic specimens of *T. minutissimus* have been available until our recent collection. *Hanken (1983a* and subsequent papers) misapplied the name to a group of populations that represent several species, three of which we have described herein. Salamanders identified as *T. minutissimus* in studies of cranial and appendicular morphology by *Hanken (1982)*, *Hanken (1983b)*, *Hanken (1984)* and *Hanken (1985)* instead belong to *T. longicaudus*.

Based on DNA sequence data, *Thorius minutissimus* is separated from topotypic *T. narisovalis* by a GTR distance of 0.122 for cyt *b* and 0.026 for 16S (*Rovito et al., 2013*). Comparable distances to the three other species treated in this paper (all from their respective type localities) are all larger, as follows: *T. pinicola*, 0.131 and 0.048; *T. tlaxiacus*, 0.134 and 0.044; and *T. longicaudus*, 0.142 and 0.039.

*Conservation status*: *Thorius minutissimus* is currently regarded as Critically Endangered: all known individuals are from a single locality and there is ongoing decline in the extent and quality of its forest habitat (*International Union for Conservation of Nature, 2016*; *Lamoreux, McKnight & Cabrera Hernandez, 2015*; *Parra-Olea, Wake & Hanken, 2008a*). Further attempts to find additional localities of this species and to more precisely define its full geographic range are urgently needed.

*Thorius narisovalis* *Taylor, 1940*
Oval-nostrilled Minute Salamander
Figure 3E

*Holotype*: FMNH 100089 (EHT-HMS 17859), Mexico, Oaxaca, "an elevation of about 2,600–3,000 meters on Cerro San Felipe, 15 km. north of Oaxaca," adult female, 18–22 August 1938, E. H. Taylor.

*Additional specimens examined*: All from Oaxaca, Mexico. Cerro San Felipe: MVZ 131153, 131155–56 (two specimens), 131158–59 (two specimens), 131161, 131451, 15.6 km NW (by road) La Cumbre, 17°14′22″N, 96°38′21″W, 3,130 m; MVZ 131162–63 (two specimens), 13.4 km NW (by road) La Cumbre, 17°13′41″N, 96°38′47″W, 3,110 m; MVZ 131166, 131168, 9.3 km NW (by road) La Cumbre, 17°12′39″N, 96°38′53″W, 3,050 m; MVZ 131446, 2.5 km NW of La Cumbre, 17°11′5″N, 96°37′26″W, 2,920 m; MVZ 162173, 162184–85 (two specimens), 162257, 186852–56 (five specimens), 186890–91 (two specimens), 4 km NW (by road) La Cumbre, 17°11′18″N, 96°36′04″W, 9360 ft; MVZ 182966, 182971–73 (three specimens), 186858–61 (four specimens), 15 km W (by road) La Cumbre, 17°14′9″N, 96°38′13″W, 3,185 m; MVZ 186857, 186882–89 (eight specimens), 9 km W (by road) La Cumbre, 17°12′35″N, 96°38′53″W, 3,080 m; IBH 26500, 12 km W (by road) La Cumbre, 17°10′80″N, 96°39′43″W, 3,100 m; IBH 22346, 6.6 km W (by road) La Cumbre, 2,860 m; and IBH 22833, 4.2 km W (by road) La Cumbre, 17°11′27″N, 96°37′38″W, 2,860 m. IBH 22988, 10 km NE (by road) Cuajimoloyas, 2,945 m; MVZ 182869–75 (seven specimens), 186862–73 (19 specimens), 4 km NE (by road) Cuajimoloyas, 17°8′4″N, 96°26′34″W, 3,170 m. MVZ 182975, 29.5 km SE (by road) Heroica Ciudad de Tlaxiaco on road to San Miguel, 17°06′05″N, 97°36′55″W, 3,080 m;

MVZ 272599, 29 km SSE (by road) Tlaxiaco, 17°08′12″N, 97°37′6″W, 3,010 m. MVZ 183012–27 (16 specimens), 15.5–15.7 mi W (by road) Zaachila, 16°55′25″N, 96°51′34″W, 2,590 m.

*Diagnosis*: Distinguished from other species of *Thorius* by the following combination of characters: (1) large size (SL exceeds 22 mm in males and 26 mm in females); (2) moderately short limbs; (3) long tail; (4) oval nostrils; (5) no maxillary teeth; (6) few vomerine teeth (fewer than 7 in both males and females); (7) unspotted belly; and (8) modal phalangeal formula in the hind limb, 1-2-3-3-1.

*Comparisons*: Adult *Thorius narisovalis* are larger than *T. arboreus, T. insperatus* and *T. papaloae*. The smallest-known adult *T. narisovalis* is 22.2 mm SL and most, especially females, are larger than 25 mm. None of the other species is known to exceed 21.4 mm and most adults, especially males, are smaller than 20 mm. *Thorius magnipes* and *T. macdougalli* have much longer limbs; LI typically is less than four. LI is 4.5 or more in *T. narisovalis*. *Thorius minutissimus* typically have relatively short tails that are as long as or shorter than SL. TL exceeds SL in all *T. narisovalis*. The nostril in *T. narisovalis* is small-to-moderate-sized and oval (occasionally round), whereas *T. pinicola* and *T. papaloae* have large and elliptical nostrils. All *T. narisovalis* lack maxillary teeth, which differentiates them from *T. adelos, T. aureus, T. schmidti* and *T. smithi*, which have maxillary teeth as adults. *Thorius narisovalis* has fewer vomerine teeth (mean number in both males and females is between four and five) than *T. longicaudus* (mean between seven and eight) and *T. boreas* (mean between nine and 10). The species is differentiated from *T. pulmonaris* and *T. tlaxiacus* in having oval rather than prolate nostrils. It is larger than *T. troglodytes* and has oval rather than elliptical nostrils.

*Variation*: Mean adult SL 25.2 mm (range 22.2–28.4) in 10 males, 27.8 mm (26.3–29.9) in 10 females. Head moderately broad; SL 7.6 times head width (6.5–8.8) in males, 7.9 (7.4–8.3) in females. Snout rounded. Nostril moderate-sized, oval (occasionally round, rarely elliptical); ratio of major to minor axes 1.4 (1.0–2.0) in both males and females. Eyes moderately large, protrude slightly beyond jaw margins in dorsal view. Suborbital groove intersects lip on each side of head. Premaxillary teeth 1.2 (0–3) in adult males, 0.5 (0–2) in females. Maxillary teeth absent. Vomerine teeth 4.3 (2–7) in males, 4.7 (3–7) in females. Limbs moderately long; LI 5.0 costal interspaces (4.5–5.5) in males, 5.7 (5.0–6.5) in females. Manus and pes relatively well developed and moderately broad; pes width 1.3 mm (1.2–1.5) in males, 1.3 mm (1.2–1.5) in females. Digits 1 and 4 (manus) and 1 and 5 (pes) short; central digits relatively long, with rounded tips. Fingers, in order of decreasing length, 3-2-4-1; toes 3-(2-4)-(1-5). Tail moderately long and tapered; SL divided by TL 0.85 (0.74–0.90) in five males, 0.82 (0.73–0.96) in eight females. Mental gland round and prominent in most adult males (maximum dimensions: 1.4 mm wide, 1.3 mm long). Postiliac gland generally pale and inconspicuous. Parotoid glands prominent in some specimens, less distinct in others.

*Coloration in life*: Dorsal stripe typically brick red or tan-brown, occasionally melanistic; venter dark, lacking conspicuous white flecks in most specimens (J. Hanken field notes, 14 July 1976; Fig. 4H). Overall color dark brownish black to black without white

spots of any size; iris black; limb insertions black (gravid female, 30 mm; D. Wake field notes, 7 October 1997; Fig. 4G).

*Coloration in preservative*: Ground color on dorsal surfaces of head, flanks and tail dark blackish brown (Fig. 3D). Prominent reddish-brown dorsal stripe from nape to anterior portion of tail in most specimens; stripe dark and inconspicuous in some individuals. Venter much paler than flanks. Numerous white spots in gular region and lower flanks in some individuals, but belly is immaculate. Pale nuchal spot in most specimens; pale area over each shoulder in some specimens. Mental gland prominent in some adult males.

*Osteology*: Skull relatively well ossified, especially in females (Figs. 5M–5O). As in other species (see above), the degree of contact between ascending processes of the premaxilla is highly variable. However, there is a greater tendency for the processes to articulate or fuse in *T. narisovalis*. Processes remain separate in relatively few specimens (character 1, state a). They articulate to varying degrees in remaining specimens (states b and c) and are fused in most males (state d). The dental process of the premaxilla is separate from the maxilla in most males (character 2, state a). However, the two elements overlap in ventral view but do not articulate in the few remaining males and in all females (states b and c). The premaxilla bears teeth in most males (character 8, state b), but teeth are absent in most females (state a). The nasal bone is thin and rod-like (character 3, state b) or slightly broader and extending somewhat anteriorly over the cartilaginous nasal capsule (state c). The nasal and maxilla are separate (character 4, state a) or barely articulate (state b). The prefrontal bone is separate from the nasal in nearly all males but in only a few females (character 5, state b). The two bones articulate (state c) or fuse (state d) in remaining specimens. The prefrontal is either separate from the maxilla (character 6, state a) or extends posteriorly beyond the nasolacrimal foramen to articulate with the maxilla (state b). The prefrontal encloses the orbitonasal foramen in a single specimen. The septomaxilla is present as a tiny sliver of bone at the edge of the external naris on one or both sides of two females (character 7, state b), but is absent in all other specimens (state a).

The maxilla is delicate—long and slender—and tapers to a point posteriorly. It lacks teeth in all specimens (character 9, state a). The vomer is moderately well developed. It includes a tiny preorbital process and bears relatively few teeth, which are arranged in a short transverse or diagonal row. The frontal fontanelle is relatively narrow. The parietal fontanelle is very broad; its mean breadth equals 0.55 times the maximum skull width across the parietals in males (range 0.42–0.69), 0.51 (0.44–0.58) in females. There is no crest on the occipito-otic and no columellar process on the operculum. The postsquamosal process is well developed. Hyobranchial cartilages are not mineralized.

There are 14 presacral vertebrae. Typically, all trunk vertebrae but the last bear ribs, but in several specimens the last trunk vertebra has a partial rib on one or both sides. Mean number of caudal vertebrae 31.3 (range 29–35) in three males, 36.8 (31–36) in five females (Fig. 8).

Limbs are slender but well developed. The tibial spur is well developed. It is present as an attached crest in most specimens but is occasionally free.

Mesopodial morphology is moderately variable. The predominant carpal pattern is I (95%; Fig. 6A); pattern III is a rare variant (5%). The modal tarsal pattern is I (85%; Fig. 6B); patterns II and V each occur at low frequency (5% and 8%, respectively). One abnormal pes has only four toes and a tarsal pattern that resembles carpal pattern I. The digital skeleton is highly variable, especially in the hind limb. The predominant phalangeal formula in the hand is 1-2-3-2 (85%); 1-2-3-1 occurs at low frequency (8%), and 1-2-2-1, 1-2-3-3 and 1-2-2-2 are rare (3% each). The modal formula in the foot is 1-2-3-3-1 (71%); 1-2-3-3-2 is a common variant (17%), and 1-2-3-2-1, 0-2-3-3-1 and 1-2-2-1 are rare (3–6% each). Limb bone epiphyses and mesopodial elements are mineralized in some adults.

*Distribution and ecology*: *Thorius narisovalis* has the largest documented geographic range of any species in the genus (Figs. 1, 7E and 7F). It is known from several mountain ranges in central and western Oaxaca, including the Sierra Aloapaneca (Cerro San Felipe and Cuajimoloyas), the Sierra de Cuatro Venados (west of Zaachila) and the Sierra de Coicoyán (southeast of Tlaxiaco; *Hanken, 1983a*). The species is confined to pine-oak forest at upper elevations, ranging from 2,780 to 3,185 m. Some localities are dominated by pine, others by oak and madrone. *Thorius* typically was more abundant in disturbed areas bearing many exposed cover objects than in mature forest.

On Cerro San Felipe, *Thorius narisovalis* occurs in sympatry with an unnamed congener (*T.* sp. 2—*Rovito et al., 2013*) and approaches a third species, *T. pulmonaris*, which occurs at lower elevations on the same mountain (*Taylor, 1940*; *Hanken, 1983a*). Other sympatric plethodontid salamanders are *Pseudoeurycea smithi* and *P. unguidentis*, as well as an unnamed species of *Chiropterotriton* (species "K"—*Darda, 1994*; *Parra-Olea, 2003*), and there are records of *P. cochranae* from this region as well. *Isthmura boneti*, which is known from nearby localities, may also occur on Cerro San Felipe, based on reports of local residents. Near Tlaxiaco, in western Oaxaca, *Thorius narisovalis* is sympatric with *T. tlaxiacus*, whereas west of Zaachila, in central Oaxaca, it occurs in sympatry with yet another unnamed congener (*T.* sp. 3—*Rovito et al., 2013*) and with *Isthmura boneti* and *Pseudoeurycea cochranae*.

*Thorius narisovalis* was very abundant historically—there are large collections of this species in MVZ, KU, LACM, UMMZ, AMNH, NMNH and other museums—but populations have declined dramatically in recent years. In October 1997, only a single live specimen was observed on Cerro San Felipe, where the species was previously extremely abundant (*Parra-Olea, García-París & Wake, 1999*; *Rovito et al., 2009*). Five more recent visits have similarly observed very few specimens: 1 on 16 August 2008, 16 on 17 March 2010, 1 on 28 June 2014, 1 on 8 August 2015, and 2 on 10 August 2015 (S. M. Rovito, 2016, unpublished data).

*Remarks*: This is one of the largest species of *Thorius*; some adults exceed 32 mm SL (*Gehlbach, 1959*). Evolutionary consequences of miniaturization of adult body size for cranial and appendicular morphology were examined by *Hanken (1982)*, *Hanken (1983b)*, *Hanken (1984)* and *Hanken (1985)*. Extensive variation in the digital skeleton includes several instances of phalangeal loss or gain. Dental polymorphism involving presence/absence of maxillary teeth among adults has been reported previously for

*T. grandis* and *T. omiltemi* (*Hanken, Wake & Freeman, 1999*). Dental polymorphism in *T. narisovalis*, involving presence/absence of premaxillary teeth, is reported here for the first time. Genetic variation was examined using protein electrophoresis by *Hanken (1980)* and *Hanken (1983a)*; populations 46–51). Based on DNA sequence data, *Thorius narisovalis* is separated from topotypic *T. pinicola* by a GTR distance of 0.108 for cyt *b* and 0.045 for 16S (*Rovito et al., 2013*). Comparable distances to the three other species treated above (all from their respective type localities) are as follows: *T. longicaudus*, 0.116 and 0.038; *T. tlaxiacus*, 0.118 and 0.043; and *T. minutissimus*, 0.122 and 0.026.

***Conservation status:*** *Thorius narisovalis* is currently regarded as Critically Endangered for a variety of reasons: drastic population declines—in excess of 80%—over the last 30–40 years; its Extent of Occurrence is probably less than 100 km$^2$ and its geographic distribution is severely fragmented (see above, Distribution and Ecology); and there is continuing decline in the extent and quality of its forest habitat (*International Union for Conservation of Nature, 2016*; *Parra-Olea, Wake & Hanken, 2008b*).

## DISCUSSION

The species considered in this paper are morphologically cryptic. Hence, molecular data have proven essential for differentiating taxa. The additional key finding that enabled us to untangle the taxonomy of populations in southern Oaxaca was our rediscovery of topotypic *T. minutissimus*. The recently published molecular phylogeny for *Thorius* showed that *T. minutissimus* is both the sister taxon of *T. narisovalis* and well differentiated from *T. longicaudus, T. pinicola* and *T. tlaxiacus* (*Rovito et al., 2013*). *Thorius narisovalis* and *T. minutissimus* form a clade with *T. boreas* and *T. aureus*, sister taxa that are sympatric in northern Oaxaca. The three southern Oaxacan species described herein belong to a separate but even larger clade that also includes three additional named species from northern Oaxaca (*T. arboreus, T. papaloae* and *T. macdougalli),* three unnamed species from northern and central Oaxaca (*Thorius* sp. 2, sp. 3 and sp. 7—*Rovito et al., 2013*) and two species from Guerrero (*T. grandis* and *T. omiltemi*). There are no molecular data for two additional named species from Guerrero, *T. infernalis* (*Hanken, Wake & Freeman, 1999*) and *T. hankeni* (*Campbell et al., 2014*). The above two clades together contain all species known from southern Oaxaca, one of which was assigned to *T. minutissimus* by *Hanken (1983a)*, who lacked topotypic material of that species. The three remaining named species from Oaxaca (*T. adelos, T. insperatus* and *T. smithi*) are even more distant relatives and clearly distinct from all the above taxa (*Rovito et al., 2013*). With our expanded taxonomic sampling and additional molecular data, we have been able to gain a better understanding of the complex nature of *Thorius* in this region and to more fully diagnose species. As noted earlier, while most species considered here are allopatric, *T. longicaudus* and *T. tlaxiacus* occur in sympatry in the vicinity of San Vicente Lachixio and *T. tlaxiacus* is sympatric with *T. narisovalis* near Tlaxiaco, two localities in southwestern Oaxaca.

*Hanken (1983a)* thorough allozymic study, using large samples, shows that *T. pinicola, T. tlaxiacus*, and *T. longicaudus* are well differentiated with respect to proteins. For example, *T. pinicola* and *T. longicaudus* show fixed allozymic differences in 4 of 18 proteins, *T. pinicola* and *T. tlaxiacus* (including both known populations) in 3 of 18 proteins, and

*T. tlaxiacus* and *T. longicaudus* (including both known populations) in 3 of 18 proteins. The number of fixed differences between *T. tlaxiacus* and *T. longicaudus* increases to five when one compares only populations from their respective type localities. Taking into account both allozymic and DNA sequence data, we are confident that six distinct named species of *Thorius*—the three we describe here as new, plus *T. minutissimus*, *T. narisovalis* and *T. pulmonaris*—occur in western and southern Oaxaca, with sympatry between *T. narisovalis* and *T. tlaxiacus* south of Tlaxiaco, between *T. tlaxiacus* and *T. longicaudus* near San Vicente Lachixio, between *T. narisovalis* and an unnamed seventh species in the Sierra de Cuatro Venados, and between *T. narisovalis*, *T. pulmonaris* and an unnamed eighth species on Cerro San Felipe (Fig. 1).

Most species of *Thorius* are difficult to distinguish from one another solely on morphological grounds, at least in part because they are so small. Moreover, taxonomic characters that reliably distinguish species in many other plethodontid genera, such as presence or absence of maxillary teeth, can be difficult to use in *Thorius* because of significant intrapopulational polymorphism (*Hanken, 1982*; *Hanken, 1984*; *Hanken, Wake & Freeman, 1999*). Nevertheless, the species do differ morphologically (*Rovito et al., 2013*). The three new species we describe here resemble one another in size, coloration and structure of the limbs and digits; all lack maxillary teeth. There are, however, subtle morphological differences, as is evident from the high percentage of specimens correctly assigned to species in the DFA (Fig. 2). Two external traits, adult body size and nostril shape, can be particularly effective at differentiating species (*Rovito et al., 2013*, Table 2). Adult *T. tlaxiacus*, for example, differ from both *T. pinicola* and *T. longicaudus* in being "very large" for the genus (> 27 mm SL, versus "large," 25–27 mm) and in having extremely elongated, prolate nostrils (vs. "elongated elliptical"). The other two species, however, are morphologically cryptic: it is extremely difficult to differentiate them externally. The numerous instances of sympatry involving two or three species of *Thorius* (*Hanken, 1983a*; *Hanken & Wake, 1994*; *Hanken & Wake, 1998*; see below) raise important questions regarding the extent of geographic variation in these species and the possibility of character displacement enabling their coexistence. We are, however, unable to address these and related questions given the limited data available, and the precarious conservation status of these taxa makes future studies unlikely. Most named species of *Thorius* are known only from their respective type localities or adjacent sites, and this is true of both *T. minutissimus* and *T. pinicola* (Fig. 1). *Thorius longicaudus* and *T. tlaxiacus* display somewhat larger ranges; each is known from two distinct localities. *Thorius narisovalis*, however, is known from at least four geographically distant localities, which define a range that spans a distance of about 150 km in central and western Oaxaca. This gives *T. narisovalis* the broadest geographic range yet demonstrated for any species in the genus. *Thorius narisovalis* also shows a complex pattern of sympatry and coexistence both with congeners and with other plethodontid genera, although several other species of *Thorius* we consider here also occur near or sympatric with one or more additional plethodontid salamanders. In some cases, a restricted range and absence of sympatric associates may accurately reflect limited geographic distribution of the species involved. In others, they may represent artifacts of the difficulties inherent in sampling

remote and inaccessible montane localities, complicated by the severe decline in population densities that we have witnessed in the past decade.

The molecular phylogenetic analysis of *Rovito et al. (2013)* provides limited insights into the relationships of the three new species we describe here. The species tree for *Thorius*, based on combined mitochondrial and nuclear gene DNA sequence data, shows the new species together within a well-supported clade that includes several other Oaxacan taxa, three of which are unnamed, as well as two Guerreran species, *T. omiltemi* and *T. grandis* (*Rovito et al., 2013*; Fig. 2, *Thorius* sp. 4, 5 and 6). And while the two populations of *T. longicaudus* appear to be closely related based on mitochondrial data alone, they are not monophyletic, whereas they are monophyletic based only on nuclear sequences. The opposite is true for the two populations of *T. tlaxiacus*. Overall, these relationships are poorly resolved, and we are able to conclude only that the three new species are close relatives.

*Thorius* may be the most endangered genus of amphibians in the world. Of the 24 named species whose conservation status has been formally evaluated (*International Union for Conservation of Nature, 2016*), 11 are listed as Critically Endangered and 12 as Endangered; the one remaining species is considered Vulnerable. The three new species described here only exacerbate this problem; we suggest that all three are also Critically Endangered. In the absence of dramatic steps to address ongoing habitat destruction, climate change, pollution and other factors that are likely contributing to calamitous population declines of neotropical plethodontid salamanders (*Rovito et al., 2009*), there is a realistic chance that the entire genus may be extinct before the end of this century. Paradoxically, at least three additional candidate species remain to be described (*Rovito et al., 2013*), although their circumstances in nature may be equally if not more perilous than those formally assessed for named species. The full extent of amphibian taxonomic diversity remains to be documented on a global scale, at the same time that populations are declining precipitously (*Hanken, 1999*).

## ACKNOWLEDGEMENTS

We thank D. Eakins, T. Hsieh and H. Shaffer for assistance in the field. At Harvard, S. Walker prepared the distribution map, J. Martinez prepared the digital X-rays and J. Woodward assisted with specimen photographs. The University of Texas High-Resolution X-ray CT Facility performed the µCT scans. L. Márquez (Laboratorio de Biología Molecular, I-Biología, UNAM) contributed additional technical assistance and support. Access to specimens and information was provided by K. Beaman (LACM), C. Conroy and B. Stein (MVZ), D. Dickey and D. Frost (AMNH), O. Flores-Villela (MZFC), J. Giermakowski (MSB), A. Resetar (FMNH), J. Rosado (MCZ) and J. Simmons (KU).

### Funding

Research was supported by grants from the Programa de Apoyo a Proyectos de Investigación e Innovación Tecnológica (PAPIIT-UNAM) IN209914, Mexico, to GP-O, and from the U.S. National Science Foundation (EF-0334846 to JH, EF-0334939 to D.B.W, and DEB-0613802 to J. Campbell); by the Council on Research and Creative Work,

University of Colorado at Boulder, and the Museum of Vertebrate Zoology, the Center for Latin American Studies, and Sigma Xi (Alpha chapter), University of California, Berkeley; and by the Putnam Expeditionary Fund of the Museum of Comparative Zoology and the David Rockefeller Center for Latin American Studies, Harvard University. The funders had no role in study design, data collection and analysis, decision to publish, or preparation of the manuscript.

## Grant Disclosures

The following grant information was disclosed by the authors:
Proyectos de Investigación e Innovación Tecnológica (PAPIIT-UNAM): IN209914.
National Science Foundation: EF-0334846, EF-0334939, DEB-0613802.
Council on Research and Creative Work, University of Colorado.
Museum of Vertebrate Zoology, the Center for Latin American Studies.
University of California, Berkeley.
Putnam Expeditionary Fund of the Museum of Comparative Zoology.
David Rockefeller Center for Latin American Studies, Harvard University.

## Competing Interests

The authors declare that they have no competing interests.

## Author Contributions

- Gabriela Parra-Olea conceived and designed the experiments, performed the experiments, analyzed the data, wrote the paper, prepared figures and/or tables, reviewed drafts of the paper.
- Sean M. Rovito conceived and designed the experiments, performed the experiments, analyzed the data, contributed reagents/materials/analysis tools, wrote the paper, prepared figures and/or tables, reviewed drafts of the paper.
- Mario García-París conceived and designed the experiments, performed the experiments, analyzed the data, wrote the paper, prepared figures and/or tables, reviewed drafts of the paper.
- Jessica A. Maisano contributed reagents/materials/analysis tools, wrote the paper, prepared figures and/or tables, reviewed drafts of the paper, digital movies of CT scans.
- David B. Wake conceived and designed the experiments, performed the experiments, analyzed the data, wrote the paper, prepared figures and/or tables, reviewed drafts of the paper.
- James Hanken conceived and designed the experiments, performed the experiments, analyzed the data, wrote the paper, prepared figures and/or tables, reviewed drafts of the paper.

## Animal Ethics

The following information was supplied relating to ethical approvals (i.e., approving body and any reference numbers):

Animal use was approved by the University of California, Berkeley, IACUC protocol #R093-0205 to DBW.

## Field Study Permissions

The following information was supplied relating to field study approvals (i.e., approving body and any reference numbers):

Collection of live salamanders in the field was authorized by the Secretaria de Recursos Naturales y del Medio Ambiente (SEMARNAT), Mexico, permit no. FAUT–0106, issued to Gabriela Parra-Olea.

## Data Deposition

Digital Morphology project, University of Texas, Austin (www.DigiMorph.org); *Thorius tlaxiacus*—DOI 10.7910/DVN/QS9S35, *Thorius pinicola*—DOI 10.7910/DVN/UTQQ3U, *Thorius narisovalis*—DOI 10.7910/DVN/HJ0IHZ, *Thorius minutissimus*—DOI 10.7910/DVN/SCPWPG, *Thorius longicaudus*—DOI 10.7910/DVN/ABO31V.

## New Species Registration

The following information was supplied regarding the registration of a newly described species:

*Thorius pinicola*
urn:lsid:zoobank.org:act:0513EF64-D4B0-4AA8-998B-AA5C2CAAE1D6.
*Thorius longicaudus*
urn:lsid:zoobank.org:act:65372EBE-28E4-4479-81D9-0477793B674E.
*Thorius tlaxiacus*
urn:lsid:zoobank.org:act:5798B848-58D7-494D-B95A-9FDA7C0D2758.
Publication
urn:lsid:zoobank.org:pub:83638F13-8A23-40F1-9992-100246084196.

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
