# Peer review of "Biology of tiny animals: three new species of minute salamanders (Plethodontidae: Thorius) from Oaxaca, Mexico"

_PeerJ, doi:10.7717/peerj.2694_

## Round 0.1 · original submission · Minor Revisions

Dear Authors,

Thank you for submitting a fine MS to PeerJ. Two of the referees suggested minor revision, while the third referee suggested accept. I have opted for minor revision so that you have the opportunity to implement the reviewers’ suggestions.

In addition to the comments of the reviewers 1 and 2, I would like to see a conservation assessment of each of the newly described species plus Thorius minutissimus and Thorius narisovalis, and a final short section on conservation species of the genus as a whole (or at least species from southern Oxaca). This should not be too much additional work, and although the scope of this MS is taxonomy, the implications of these descriptions and redescriptions together with the field collecting trends and landscape changes are profound for these species. Few people have such extensive and long-term data, thus I think there is also a certain amount of social responsibility of interpreting these data from a conservation perspective and making this information available in a clear form.

Sincerely,

Tomas Hrbek

·

Basic reporting

This manuscript is well-written and includes the relevant background information and literature, all appropriately referenced.
See my later comments regarding Figure 1 and the legend for Figure 1.

Experimental design

The study is well-designed and contains a variety of evidence, with excellent figures, analyses and discussion of the validity of the species and the evidence for the new species in the context of previous work and new evidence.

Validity of the findings

I think the declines or possible extinction needs to be emphasized more. Either in the introduction or the abstract, you could mention the proportion of material collected long ago --
Something like "[Nearly all or most or X%] of the specimens analyzed in this study were collected more than X years ago because living specimens have become nearly impossible to find"
Why exclude sp. 2 and sp. 3? I think the paper is fine as it is, but in the conclusions somewhere it would be useful to stress the paradox presented here of the 3 new species presented here, plus [4?] additional ones to be described soon increasing the taxonomic diversity while the species in the wild are in peril.

sp. 4 (longicauda) is not monophyletic in the mtDNA tree in Rovito et al, 2013 - this is mentioned in the text, but I would like to see that the morphology between the 2 localities is not separate in morphospace (Figure 2) - if the 2 populations are discrete, then it needs to be mentioned, if not, that fact should be mentioned.

Figure 1 - What is white versus black symbols inside of the symbols?- this needs to be clear in Figure legend.
The inset is too enlarged to be useful for someone not familiar with Mexican geography.

Is the mintissimus locality the same one as the 2013 one? - They don't match up.

In addition, the legend mentions that T. pulmonaris is not shown, but sp. 2 and sp. 3, and possibly sp. 7 are not shown, but occur on the map. This is important because of the discussion of sympatry and the fact that the map as it is suggests a lack of sympatry for many samples, but sp. 2, and sp. 3 are sympatric with narisovalis.

Figure 2: More discussion and examination of sympatry is needed in the paper:
If character displacement occurs for sympatric species, then the prediction would be that allopatric species can overlap in morphospace, but sympatric forms should be divergent.
Is that true at all? If so, it needs to be mentioned.
T. pinicola overlaps with other species, but is not sympatric with any of them, as would be expected with character displacement.
For the two individuals of longicaudus and tlaxiacus that are similar to each other in morphospace - are these allopatric or sympatric? The same question for narisovalis and tlaxiacus apply.
Even if there is no pattern, this lack of pattern should be mentioned.

Additional comments

The only two changes I see as necessary are:
1) Improve and clarify Figure 1 and
2) an examination and discussion of sympatry as it relates to the individuals shown in Figure 2

Reviewer 2 ·

Basic reporting

Parra - Olea et al. described three new species of Thorius and redescribed two species. The article is well written and I think that the three new salamander species described are new.

The evolutionary characteristic used by the authors to distinguish the new species as well as the diagnostic features and comparisons with other species are well done.

The title: I think it would need to change to be more specific (see comments),
the principal reason of my suggestion is beacuse the authors not access to review all species of the genus under study to said: Taxonomy of minute salamanders (thats mean that you are gonna revised all the species of the genus).

Although based largely on the work of Rovito et al. 2013 for the molecular references, the authors do not report the genetic distances (it would be interesting as evidence) to differentiate species. Also is important report the support the branches obtained by Rovito et al. 2013 in each clade of the species and report which genes were used.

i detected some imprecise terms that need to be improve: e.g. moderate (without mention of measures); long tail (without measure), genetic distances was 0.15 (that mean high or low for the group?; etc..(see the comments)

In addition to the background distribution mentioned by the authors, is necessary to highlight the importance of conservation of these small
organisms. Add a sub part of conservation in each described species would be important to draw the attention of readers.

some paragraphs in the discussion not discuss the results, but rather only tell the taxonomic history of the species ( that should be in redescriptions as remarks).

Experimental design

The authors use morphologic, morphometric, phylogenetic and Osteological characteres to describe the new species.

I think that the experimental design are very complete and the evolutionary tools that were used are efficient and informative

Validity of the findings

I consider this work important to reveal the hidden species diversity within the genus.

Additional comments

No comments

Reviewer 3 ·

Basic reporting

Well done.

Experimental design

Well done.

Validity of the findings

Well done.

Additional comments

This manuscript provides a very thorough and detailed description of three new species of plethodontid salamander from the genus Thorius, and clarifies the taxonomy and distribution of a fourth species from which the other species are split and described. The authors provide substantial morphological information for these species descriptions, which are backed up by sufficient genetic data. The strength of the genetic evidence comes from previously published allozyme data, which even in an era of genomic sequence data, have substantial information about cryptic species boundaries. The DNA sequence evidence used here is a bit more limited, but is largely in line with the allozyme-based insights. This paper is very well written, especially given the level of detail provided in these descriptions. How many more species of plethodontid salamanders are there to describe? Clearly at least three more.

---

## Round 0.2 · accepted · Accept

Dear Authors,

Thank you for making revisions to your fine paper. I appreciate that you included the conservation evaluation for the species treated in this paper. It is timely, and hopefully it will have a positive impact on the conservation of this enigmatic groups of salamanders.

Congratulations!

Tomas Hrbek

---

## Author Rebuttal · Round 0.2

# MUSEUM OF COMPARATIVE ZOOLOGY
*The Agassiz Museum*

[Figure]

HARVARD UNIVERSITY
26 OXFORD STREET
CAMBRIDGE, MASSACHUSETTS 02138

10 October 2016

Editorial Office
*PeerJ*, Inc.
PO Box 910224
San Diego, CA 92191

RE: 2016:07:12251:0:1:NEW 28 Jul 2016, Parra-Olea et al., "Taxonomy of minute
    salamanders…"

Dear Editors:

On behalf of my coauthors and myself, I thank Academic Editor Tomas Hrbek and the three reviewers for their very perceptive and helpful comments on the above manuscript. We have addressed all their recommendations and other comments in the revised manuscript, which we believe is now suitable for publication in *PeerJ*.

The following pages summarize our actions taken in response to each comment. Please note that some of reviewer 1's suggestions required revision of Figure 1; the revised illustration has been uploaded, along with publication-quality versions of the other figures.

I also want to thank *The PeerJ Team* for granting additional time for us to complete the revision and have all authors review and approve the numerous edits.

Sincerely yours,

James Hanken
Alexander Agassiz Professor of Zoology,
Curator in Herpetology, and Director

**Editor's Comments**

**MINOR REVISIONS**

**Dear Authors,**

**Thank you for submitting a fine MS to PeerJ. Two of the referees suggested minor revision, while the third referee suggested accept. I have opted for minor revision so that you have the opportunity to implement the reviewers' suggestions.**

**In addition to the comments of the reviewers 1 and 2, I would like to see a conservation assessment of each of the newly described species plus Thorius minutissimus and Thorius narisovalis, and a final short section on conservation species of the genus as a whole (or at least species from southern Oxaca). This should not be too much additional work, and although the scope of this MS is taxonomy, the implications of these descriptions and redescriptions together with the field collecting trends and landscape changes are profound for these species. Few people have such extensive and long-term data, thus I think there is also a certain amount of social responsibility of interpreting these data from a conservation perspective and making this information available in a clear form.**

**Sincerely,**

**Tomas Hrbek**

We have added a conservation assessment to each species account and a summary assessment at the end of the Discussion.

**Reviewer 1 (Todd Jackman)**

## Basic reporting

**This manuscript is well-written and includes the relevant background information and literature, all appropriately referenced. See my later comments regarding Figure 1 and the legend for Figure 1.**

## Experimental design

**The study is well-designed and contains a variety of evidence, with excellent figures, analyses and discussion of the validity of the species and the evidence for the new species in the context of previous work and new evidence.**

## Validity of the findings

**I think the declines or possible extinction needs to be emphasized more. Either in the**

**introduction or the abstract, you could mention the proportion of material collected long ago --**
**Something like "[Nearly all or most or X%] of the specimens analyzed in this study were collected more than X years ago because living specimens have become nearly impossible to find"**

We added short statements to the Abstract and Introduction to emphasize population declines and extinction risks.

**Why exclude sp. 2 and sp. 3? I think the paper is fine as it is, but in the conclusions somewhere it would be useful to stress the paradox presented here of the 3 new species presented here, plus [4?] additional ones to be described soon increasing the taxonomic diversity while the species in the wild are in peril.**

We added a paragraph regarding conservation threats at the end of the Discussion. The last two sentences address the paradox noted in R1's comment above.

**sp. 4 (longicauda) is not monophyletic in the mtDNA tree in Rovito et al, 2013 - this is mentioned in the text, but I would like to see that the morphology between the 2 localities is not separate in morphospace (Figure 2) - if the 2 populations are discrete, then it needs to be mentioned, if not, that fact should be mentioned.**

Each species depicted in Figure 2 is represented by specimens only from its respective type locality, so the separation among specimens of *T. longicaudus* represents intrapopulational and not geographic variation within this species. We have revised corresponding sections of the Materials & Methods, Results, Tables 1 and 2, and Figure 2 to specify locality information.

**Figure 1 - What is white versus black symbols inside of the symbols?- this needs to be clear in Figure legend.**

We revised both the map and its legend to make the meaning of these symbols more explicit.

**The inset is too enlarged to be useful for someone not familiar with Mexican geography.**

The inset has been revised to include more of southern Mexico and help the reader understand the region of interest in this study.

**Is the mintissimus locality the same one as the 2013 one? - They don't match up.**

The locality in Rovito et al. (2013) is correct, and we have revised the map in the present manuscript to match it. (In so doing, we discovered that IUCN's distribution map also is in error.)

**In addition, the legend mentions that T. pulmonaris is not shown, but sp. 2 and sp. 3, and possibly sp. 7 are not shown, but occur on the map. This is important because of the discussion of sympatry and the fact that the map as it is suggests a lack of sympatry for many samples, but sp. 2, and sp. 3 are sympatric with narisovalis.**

We added known localities of *T. pulmonaris* and of candidate species 2 and 3, and indicate where these species are sympatric with the five other species already depicted on the map.

**Figure 2: More discussion and examination of sympatry is needed in the paper:**
**If character displacement occurs for sympatric species, then the prediction would be that allopatric species can overlap in morphospace, but sympatric forms should be divergent.**
**Is that true at all? If so, it needs to be mentioned.**
**T. pinicola overlaps with other species, but is not sympatric with any of them, as would be expected with character displacement.**
**For the two individuals of longicaudus and tlaxiacus that are similar to each other in morphospace - are these allopatric or sympatric? The same question for narisovalis and tlaxiacus apply.**
**Even if there is no pattern, this lack of pattern should be mentioned.**

As explained above, data depicted in Figure 2 do not allow meaningful assessment of morphological variation among specimens in sympatry (vs. allopatry) because each species is represented by specimens only from its respective type locality, and all five type localities are geographically distinct. Thus, none of the specimens represented occurred in sympatry with one another. We have revised corresponding sections of the Materials & Methods, Results, Tables 1 and 2, and Figure 2 to clarify the geographic provenance of the data. In addition, we added text in Results that explains the two instances of morphological overlap between single specimens of two different species that the reviewer notes above, and we state explicitly that these data do not allow evaluation of patterns of morphological divergence in sympatry vs. allopatry. Finally, we add a sentence to the Discussion that identifies such patterns, including the possibility of character displacement, as very deserving subjects of future research in Thorius, which presents many instances of sympatry of closely related species.

## Comments for the Author

**The only two changes I see as necessary are:**
**1) Improve and clarify Figure 1 and**

**2) an examination and discussion of sympatry as it relates to the individuals shown in Figure 2**

Both necessary changes have been made, as described in greater detail above.

## Annotated manuscript

**The reviewer has also provided an annotated manuscript as part of their review:**

Annotations proved helpful as we addressed R1's suggestions above.

**Reviewer 2 (Anonymous)**

## Basic reporting

**Parra - Olea et al. described three new species of Thorius and redescribed two species. The article is well written and I think that the three new salamander species described are new.**

**The evolutionary characteristic used by the authors to distinguish the new species as well as the diagnostic features and comparisons with other species are well done.**

**The title: I think it would need to change to be more specific (see comments), the principal reason of my suggestion is beacuse the authors not access to review all species of the genus under study to said: Taxonomy of minute salamanders (thats mean that you are gonna revised all the species of the genus).**

We revised the title to avoid suggesting that we review all species of *Thorius* in Oaxaca.

**Although based largely on the work of Rovito et al. 2013 for the molecular references, the authors do not report the genetic distances (it would be interesting as evidence) to differentiate species. Also is important report the support the branches obtained by Rovito et al. 2013 in each clade of the species and report which genes were used.**

We 1) added the level of support (posterior probability value) of each clade from Rovito et al.; 2) report which genes were sequenced in that study; and 3) added GTR distances for two genes between each new species and the other species considered in this study.

**i detected some imprecise terms that need to be improve: e.g. moderate (without mention of measures); long tail (without measure), genetic distances was 0.15 (that mean high or low for the group?; etc..(see the comments)**

The above imprecise terms were defined more precisely (based on Rovito et al. 2013) and text was added to provide sufficient context to interpret the significance of a given Nei distance value.

**In addition to the background distribution mentioned by the authors, is necessary to highlight the importance of conservation of these small organisms. Add a sub part of conservation in each described species would be important to draw the attention of readers.**

As stated above (response to editor's comments), we have added a conservation assessment to each species account and a summary assessment at the end of the Discussion.

**some paragraphs in the discussion not discuss the results, but rather only tell the taxonomic history of the species ( that should be in redescriptions as remarks).**

One entire paragraph and most of a second paragraph were dropped from the Discussion and the material incorporated into the Remarks sections of corresponding species accounts.

## Experimental design

**The authors use morphologic, morphometric, phylogenetic and Osteological characteres to describe the new species.**

**I think that the experimental design are very complete and the evolutionary tools that were used are efficient and informative**

## Validity of the findings

**I consider this work important to reveal the hidden species diversity within the genus.**

## Comments for the Author

**No comments**

**Reviewer 3 (Anonymous)**

## Basic reporting

**Well done.**

## Experimental design

**Well done.**

## Validity of the findings

**Well done.**

## Comments for the Author

**This manuscript provides a very thorough and detailed description of three new species of plethodontid salamander from the genus Thorius, and clarifies the taxonomy and distribution of a fourth species from which the other species are split and described. The authors provide substantial morphological information for these species descriptions, which are backed up by sufficient genetic data. The strength of the genetic evidence comes from previously published allozyme data, which even in an era of genomic sequence data, have substantial information about cryptic species boundaries. The DNA sequence evidence used here is a bit more limited, but is largely in line with the allozyme-based insights. This paper is very well written, especially given the level of detail provided in these descriptions. How many more species of plethodontid salamanders are there to describe? Clearly at least three more.**

We thank this reviewer for his/her comments, which require no manuscript revision.

## Technical changes

**These are your technical changes from PeerJ staff:**
**# Tables**
**1) PDF is not an accepted file format for tables.**

Understood.  Tables were submitted as pdf's only for manuscript review.

**2) Please upload your tables in separate Word documents here <https://peerj.com/manuscripts/12251/files>. [note: Tables should not be a .jpg or .pdf image of a table pasted into the Word document.]**

Done.

**# Manuscript Source File**
**1) Please provide the clean unmarked source file (e.g. .DOCX, .DOC, .ODT) with no tracked changes shown, all tracked changes accepted and tracked changes turned off.**

Done.

**2) Please upload the manuscript file in the Revised Manuscript & Primary Files section here: https://peerj.com/manuscripts/12251/files/.**

Done.

**3) If you uploaded a PDF because of formatting problems, please provide the source file as a Supplemental File and we will mark it as the correct file type as necessary if the manuscript is accepted.**

n/a.

**# Figure Permissions**
**Could you please confirm that Figure 1 is not copyrighted, or if it is, you have permission to publish it under our CC BY 4.0 license? If it is taken from copyrighted material, we will need a copy of the written permission for our records. A sample permission letter can be found here: <https://peerj.com/about/author-instructions/#figure-referencing>**

Figure 1 has not been published before and is not copyrighted.  It was prepared from scratch for this manuscript by Mr. Scott Walker, Digital Cartography Specialist, Harvard Map Library.